# Cysteinyl-tRNA synthetase governs cysteine polysulfidation and mitochondrial bioenergetics

Takaaki Akaike[1], Tomoaki Ida[1], Fan-Yan Wei[2], Motohiro Nishida[3,4], Yoshito Kumagai[5], Md. Morshedul Alam[6], Hideshi Ihara[7], Tomohiro Sawa[8], Tetsuro Matsunaga[1], Shingo Kasamatsu[1], Akiyuki Nishimura[3], Masanobu Morita[1], Kazuhito Tomizawa[2], Akira Nishimura[1], Satoshi Watanabe [9], Kenji Inaba[9], Hiroshi Shima[10], Nobuhiro Tanuma[10], Minkyung Jung[1], Shigemoto Fujii[1], Yasuo Watanabe[11], Masaki Ohmuraya[12], Péter Nagy[13], Martin Feelisch[14], Jon M. Fukuto[15] & Hozumi Motohashi [6]

Cysteine hydropersulfide (CysSSH) occurs in abundant quantities in various organisms, yet little is known about its biosynthesis and physiological functions. Extensive persulfide formation is apparent in cysteine-containing proteins in *Escherichia coli* and mammalian cells and is believed to result from post-translational processes involving hydrogen sulfide-related chemistry. Here we demonstrate effective CysSSH synthesis from the substrate L-cysteine, a reaction catalyzed by prokaryotic and mammalian cysteinyl-tRNA synthetases (CARSs). Targeted disruption of the genes encoding mitochondrial CARSs in mice and human cells shows that CARSs have a crucial role in endogenous CysSSH production and suggests that these enzymes serve as the principal cysteine persulfide synthases in vivo. CARSs also catalyze co-translational cysteine polysulfidation and are involved in the regulation of mitochondrial biogenesis and bioenergetics. Investigating CARS-dependent persulfide production may thus clarify aberrant redox signaling in physiological and pathophysiological conditions, and suggest therapeutic targets based on oxidative stress and mitochondrial dysfunction.

[1] Department of Environmental Health Sciences and Molecular Toxicology, Tohoku University Graduate School of Medicine, Sendai 980-8575, Japan. [2] Department of Molecular Physiology, Graduate School of Medical Sciences, Kumamoto University, Kumamoto 860-8556, Japan. [3] Division of Cardiocirculatory Signaling, Okazaki Institute for Integrative Bioscience, National Institute for Physiological Sciences, National Institutes of Natural Sciences, Okazaki 444-8787, Japan. [4] Department of Translational Pharmaceutical Sciences, Graduate School of Pharmaceutical Sciences, Kyushu University, Fukuoka 812-8582, Japan. [5] Environmental Biology Section, Faculty of Medicine, University of Tsukuba, Tsukuba 305-8575, Japan. [6] Department of Gene Expression Regulation, Institute of Development, Aging and Cancer, Tohoku University, Sendai 980-8575, Japan. [7] Department of Biological Science, Graduate School of Science, Osaka Prefecture University, Osaka 599-8531, Japan. [8] Department of Microbiology, Graduate School of Medical Sciences, Kumamoto University, Kumamoto 860-8556, Japan. [9] Institute of Multidisciplinary Research for Advanced Materials, Tohoku University, Sendai 980-8577, Japan. [10] Division of Cancer Chemotherapy, Miyagi Cancer Center Research Institute, Natori 981-1293, Japan. [11] Laboratory of Pharmacology, Showa Pharmaceutical University, Tokyo 194-8543, Japan. [12] Department of Genetics, Hyogo College of Medicine, Nishinomiya, Hyogo 663-8501, Japan. [13] Department of Molecular Immunology and Toxicology, National Institute of Oncology, Budapest 1122, Hungary. [14] Clinical and Experimental Sciences, Faculty of Medicine, University of Southampton, Southampton General Hospital and Institute for Life Sciences, Southampton SO16 6YD, UK. [15] Department of Chemistry, Sonoma State University, Rohnert Park, CA 94928, USA. Tomoaki Ida, Fan-Yan Wei, Motohiro Nishida and Yoshito Kumagai contributed equally to this work. Correspondence and requests for materials should be addressed to T.A. (email: takaike@med.tohoku.ac.jp)

Cysteine hydropersulfide (CysSSH) is found physiologically in prokaryotes, eukaryotic cells, and mammalian tissues[1,2]. Previously, we unequivocally verified the presence of remarkable amounts of CysSSH, glutathione persulfide (GSSH), and longer chain sulfur compounds (polysulfides, including CysS/GS–(S)$_n$–H) in cultured cells and tissues in vivo in mice and humans[3–6]. The chemical properties and abundance of these species suggest a pivotal role for reactive persulfides (i.e., compounds containing an—SSH group) in cell-regulatory processes. Researchers proposed that CysSSH and related species can behave as potent antioxidants and cellular protectants, and may function as redox signaling intermediates[3–10]. Persulfides are also essential structural components of several proteins and enzymes, e.g. serving as metal ligands in iron-sulfur clusters (or sulfide donors) and in iron-cysteine and zinc-cysteine complexes[11–15]. In fact, the existence of a cell reservoir for sulfane sulfur (sulfur-bonded sulfur atoms with six electrons), including low-molecular-weight (LMW) and protein-bound cysteine polysulfides, has long been known[1,3–7,15,16]. Thus, although the prevalence of endogenous polysulfides is clearly established and their biological relevance increasingly being recognized, the chemical biology and physiological functions of these species are not known with any certainty. Current dogma holds that persulfide/polysulfide formation arises as a result of hydrogen sulfide (H$_2$S) oxidation[3,4,7–9] or chemical reaction with nitric oxide[3,17]. Two H$_2$S-generating enzymes involved in sulfur-containing amino acid metabolism—cystathionine γ-lyase (cystathionase, CSE) and cystathionine β-synthase (CBS)—can catalyze CysSSH biosynthesis using cystine (CysSSCys) as a substrate[3,4,6–10,18–21]. However, the observed $K_m$ is high, and both cells and mice lacking CSE and/or CBS still display appreciable levels of CysSSH[20–24], which suggests the possibility that alternative processes may be responsible for endogenous persulfide production. Thus, it appears that other biosynthetic routes of CysSSH formation exist that have yet to be identified.

This study reveals that cysteinyl-tRNA synthetases (CARSs), in addition to their canonical role in protein translation, act as the principal cysteine persulfide synthases (CPERSs) in vivo. CARSs play a novel and prominent role in endogenous production of both LMW polysulfides and polysulfidated proteins that are abundantly detected in cells and in mice. Notably, CARS2, a mitochondrial isoform of CARS, is involved in mitochondrial biogenesis and bioenergetics via CysSSH production.

## Results

**Redox property of cysteine and protein polysulfides.** CysSSH has unique redox-active properties that distinguishes it from the cysteine (CysSH) thiol. In evaluating the physiological rationale for biological CysSSH production, our present study confirmed that cysteine persulfide/polysulfides (CysSSH/CysS–(S)$_n$–H) possess mixed sulfur reactivity—both nucleophilic and electrophilic (Supplementary Figs 1 and 2)—a property that is unique and distinct from that of other simple biologically relevant thiols. The dual electrophilic-nucleophilic character of hydropersulfides is well documented (the anionic RSS⁻ species being nucleophilic and the protonated RSSH species possessing electrophilic properties akin to disulfides, RSSR)[25–27]. Moreover, dialkylpolysulfides can also be nucleophilic and electrophile-mediated cleavage of S-S bonds is established[28]. The unique properties and reactivity of polysulfides allowed us to develop several analytical techniques aimed at determining endogenous production of LMW and protein-bound polysulfides (Supplementary Fig. 3). We first developed a convenient method for selective detection of polysulfidated proteins: the biotin-polyethylene glycol (PEG)-conjugated maleimide (biotin-PEG-MAL) labeling gel shift assay

(PMSA; Supplementary Fig. 3a)[15]. PMSA demonstrated extensive protein-bound cysteine polysulfidation (Supplementary Fig. 4), not only for recombinant proteins, prepared in an *Escherichia coli* cell expression system (Supplementary Table 1) but also for endogenous proteins expressed in mammalian cells.

We then used liquid chromatography-electrospray ionization-tandem mass spectrometry (LC-ESI-MS/MS) with β-(4-hydroxyphenyl)ethyl iodoacetamide (HPE-IAM) as a trapping agent to identify and precisely quantify various hydropolysulfides, and also to verify the site specificity of polysulfidation as well as the number of sulfur atoms involved in proteins (Supplementary Fig. 5, and Supplementary Table 2). We chose HPE-IAM for the LC-ESI-MS/MS analyses, as described recently[6] because of its mild electrophilicity that ensures specific labeling of hydropolysulfides to form stable adducts without appreciable artifactual decay related to their dual nucleophilic and electrophilic character (Supplementary Fig. 2). In fact, we quantified CysS–(S)$_n$–H formed in alcohol dehydrogenase 5 (ADH5) and glyceraldehyde-3-phosphate dehydrogenase (GAPDH) by LC-MS/MS analysis, after pronase digestion of the HPE-IAM-labeled proteins, which revealed that more than 70% of cysteine residues were polysulfidated (Fig. 1a and Supplementary Fig. 6), a result consistent with the PMSA profile alluded to above (Supplementary Fig. 4). The treatment of ADH5 with *N*-ethylmaleimide (NEN) indeed completely abrogated the HPE-IAM labeling of CysSH and CysSSH/SSSH as evidenced by LC-ESI-MS/MS analysis shown in Supplementary Fig. 6b. This data indirectly supports the electrophilic decomposition of protein-bound cysteine polysulfides induced by a strong electrophile NEM. Additional LC-quadrupole (Q)-time-of-flight (TOF)-MS analyses identified sites of polysulfide formation and the sulfur chain length in each protein (Supplementary Fig. 7).

**Protein polysulfidation induced by cysteinyl-tRNA synthetase.** Because such extensive protein polysulfidation is unlikely to occur effectively by simple chemical means[3,4,7–10], we hypothesized that CysSSH and CysS–(S)$_n$–H may be incorporated during protein translation. To evaluate this hypothesis, we analyzed the incorporation of CysSSH/CysS–(S)$_n$–H into tRNA via cysteinyl-tRNA synthetase (CARS) from *E. coli* (EcCARS) by using synthetic CysS–(S)$_n$–H and LC-MS/MS analyses (Supplementary Fig. 8). We observed effective production of CysSSH-bound tRNA (Cys-tRNA$^{CysSSH}$), which indeed suggests translational incorporation of CysSSH/CysS–(S)$_n$–H into proteins. Unexpectedly, we identified extremely high levels (>80% of total cysteine residues) of tRNA-bound cysteine persulfide, trisulfide, and even tetrasulfide, when using simple (native) cysteine with EcCARS (Fig. 1b and Supplementary Fig. 9). As an important result, these cysteine polysulfides bound to tRNA were effectively incorporated into nascent polypeptides, which is synthesized de novo in the ribosomes (Fig. 1c), as verified by a modification of the puromycin-associated nascent chain proteomics (PUNCH-P) method[29], here termed PUNCH-PsP, PUNCH for Polysulfide Proteomics (Supplementary Fig. 10). This PUNCH-PsP analysis allowed us to obtain specific and selective identification of the intact forms of CysS–(S)$_n$–H residues in the nascent peptides of GAPDH present only within the ribosomes of *E. coli*, as Supplementary Fig. 10a shows. We clearly identified high degrees of polysulfidation occurring at the $^{247}$Cys residue of the mature GAPDH protein expressed and synthesized in *E. coli*. All native forms of CysSH, CysSSH, and CysSSSH residues were efficiently recovered from the native whole GAPDH protein and the extension of polysulfidation reached more than 60% of the $^{247}$Cys residue of mature protein (Supplementary Fig. 10e). All these rigorous LC-Q-TOF analyses unambiguously revealed that extensive and

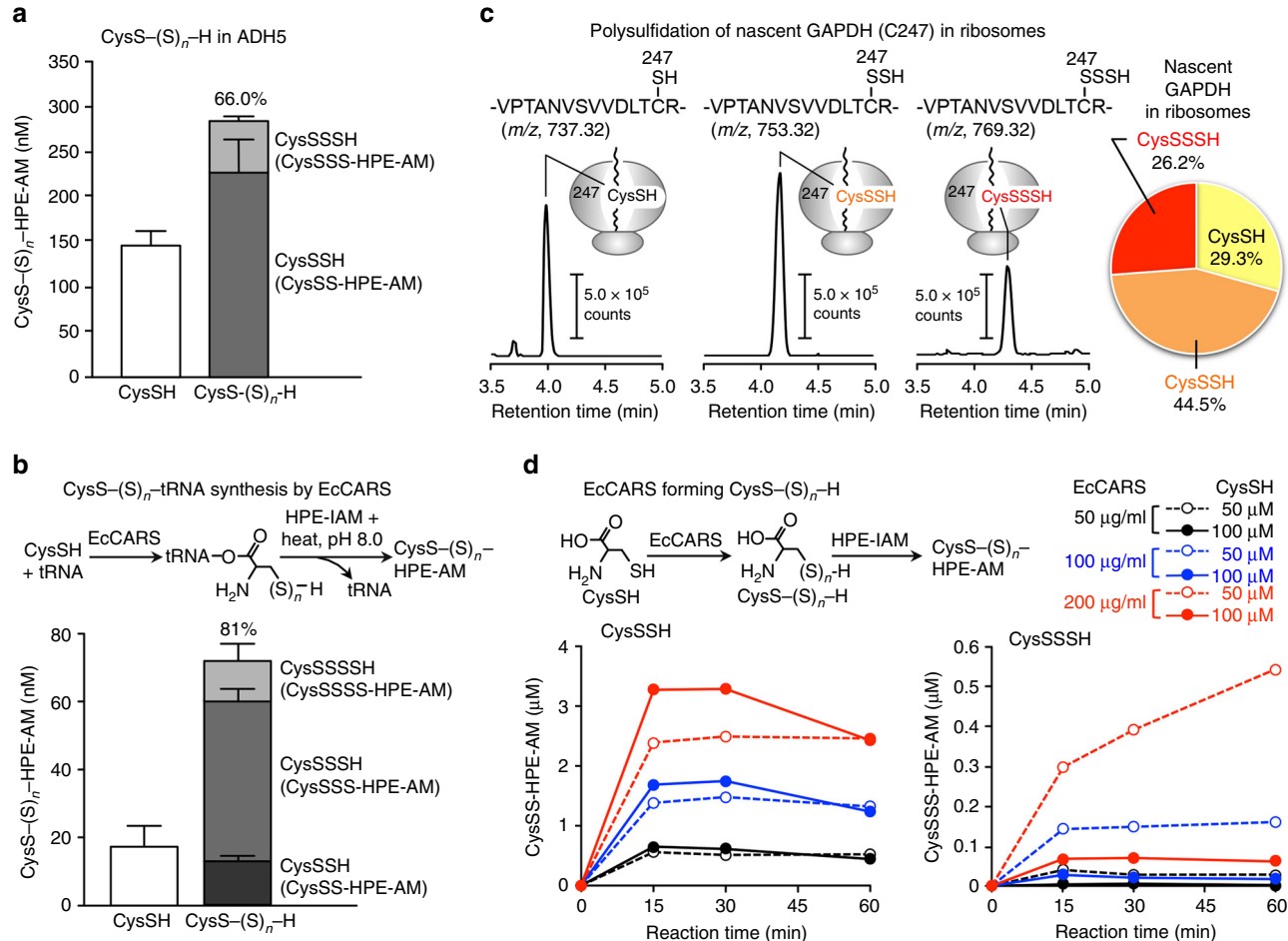

**Fig. 1** Formation of cysteine persulfide (CysSSH) and CysS–(S)$_n$–H in proteins and their biosynthesis by EcCARS. **a** Quantitative identification by LC-MS/ MS analysis of CysS–(S)$_n$–H formed in recombinant ADH5 after pronase digestion of the HPE-IAM-labeled protein. **b** Formation of cysteine (CysSH) and CysS–(S)$_n$–H on tRNA (Cys-tRNA$^{CysS-(S)n-H}$) as identified by HPE-IAM labeling LC-MS/MS analysis, which determined the amounts of CysSH and CysS– (S)$_n$–H released from Cys-tRNA$^{Cys}$ and Cys-tRNA$^{CysS-(S)n-H}$ synthesized in the EcCARS enzymatic reaction after their heat or alkaline treatment. The method employed is illustrated in the upper panel. **c** GAPDH cysteine polysulfides are formed and incorporated into nascent polypeptides synthesized de novo in ribosomes, as identified by PUNCH-PsP (Supplementary Fig. 10; cf. Supplementary Fig. 6). **d** CysS–(S)$_n$–H formation from cysteine, catalyzed by EcCARS, as dependent on enzyme and substrate (cysteine) concentrations and reaction time (lower panel). Schematic representation of the EcCARS-catalyzed reaction (upper panel). HPE-AM, β-(4-hydroxyphenyl)ethyl acetamide; HPE-IAM, β-(4-hydroxyphenyl)ethyl iodoacetamide. Data **a**, **b** are means ± s.d. (n = 3)

prevalent cysteine polysulfidation is introduced co-translationally and sustained in the mature protein physiologically present even in the post-translational processes of the cells.

Consistent with these findings, EcCARS itself appeared to have strong catalytic activity for generating CysS–(S)$_n$–H (CysSSH and CysSSSH) from the natural substrate cysteine (Fig. 1d). The persulfide synthase activity of EcCARS depended partly on added pyridoxal phosphate (PLP) (Fig. 2a) but not on ATP and tRNA: the latter two being required for Cys-tRNA$^{Cys}$ biosynthesis by EcCARS. Persulfide generation by EcCARS was enantioselective, because only L-cysteine but not D-cysteine demonstrated activity, which ruled out nonspecific post-translational persulfidation. Furthermore, we performed a stable isotope ($^{34}$S) tracer experiment combined with LC-MS/MS-based HPE-IAM assay to clarify the catalytic mechanism of cysteine polysulfidation by EcCARS (Supplementary Fig. 11). Specifically, by means of LC-MS/MS analysis for the enzymatic reaction with stable isotope ($^{34}$S)-labeled cysteine as a substrate, we found that EcCARS catalyzed the cleavage of a sulfur atom from one cysteine and its transfer to another cysteine to form CysSSH, as Supplementary Fig. 11a illustrates.

**Identification of CARSs as CPERSs**. Kinetic analyses confirmed that, because of a very low Michaelis constant $K_m$ and high catalytic rate constant $k_{cat}$, EcCARS is very efficient in producing CysSSH, i.e., functioning as a CPERS, with a high affinity for cysteine (Supplementary Fig. 12 and Supplementary Table 3), in particular when compared with the kinetic parameters of other enzymes such as CSE (Supplementary Table 3)[7,21]. Although the $k_{cat}/K_m$ value is almost equal to values of EcCARS, CSE, and CBS utilize only cystine (but not cysteine) as a substrate, which is quite distinct from CARSs that use cysteine (but not cystine) for CysSSH production[3]. In addition, because the intracellular cystine content range is physiologically at low micromolar or sub-micromolar concentrations, which are far lower than the $K_m$ value of CSE (more than 200 μM), CSE cannot directly utilize cysteine for persulfide production. Also, the cystine/CSE reaction may not compete successfully with the reactions with other enzymes metabolizing cystine and substance such as glutathione, which exists abundantly in cells and thus readily interacts with cysteine under physiological conditions. The intracellular cysteine concentration is reportedly 100–1000 μM in cells and major organs[3], which is much higher than the $K_m$ of CARS. These

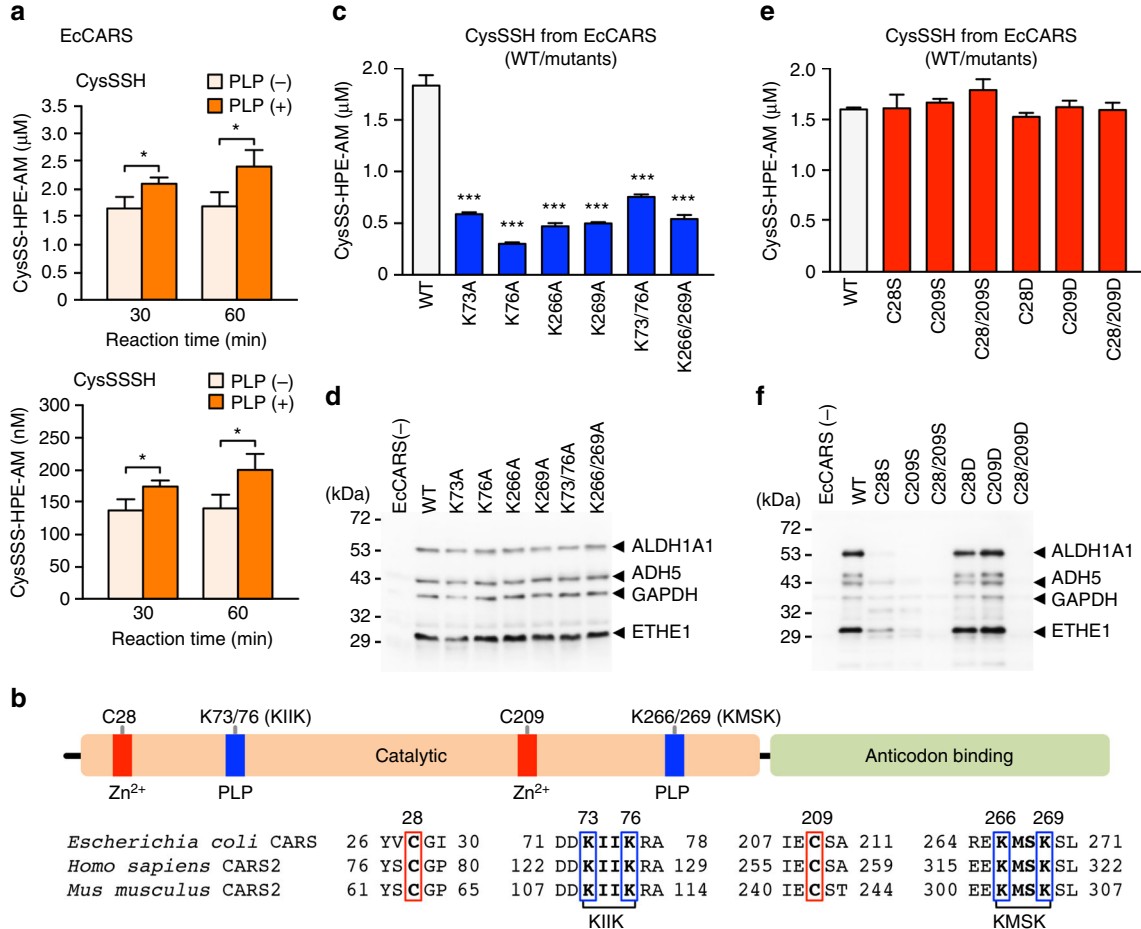

**Fig. 2** CysS–(S)$_n$–H biosynthesis catalyzed by EcCARS and its various mutant EcCARSs. **a** CysS–(S)$_n$–H (CysSSH and CysSSSH) biosynthesis from cysteine catalyzed by EcCARS as a function of reaction time and the presence or absence of PLP. CysS–(S)$_n$–H production was analyzed by using the HPE-IAM labeling with LC-MS/MS analysis for the reaction of recombinant EcCARS (200 μg/ml) with 100 μM cysteine in the presence or absence of 50 μM PLP. The data are means ± s.d. (n = 3). *P < 0.05. **b** General structure (upper panel) and conserved amino acid alignments (lower panel) of bacterial, human, and rodent CARSs. **c**, **e** Enzyme activities of EcCARS lysine (K) mutants **c** and cysteine (C) mutants **e** to form CysSSH. WT and EcCARS K and C mutants, 200 μg/ml each, reacted with 25 μM cysteine at 37 °C for 30 min. Data represent means ± s.d. (n = 3). ***P < 0.001. The enzyme activity of EcCARS Lys (**d**) and Cys (**f**) mutants was assessed by the PUREfrex assay with the cell-free translational reactions for ALDH1A1 (55 kDa), ADH5 (40 kDa), GAPDH (36 kDa), and ETHE1 (28 kDa), with protein syntheses being identified by western blotting

biochemical reports, therefore, strongly suggest that CARS can function as a major source of CysS–(S)$_n$–H generation under physiological conditions.

Investigation of EcCARS PLP-binding sites with LC-Q-TOF-MS analysis and Mascot data searches indeed revealed that lysine (K) residues, including $^{73}$KIIK$^{76}$ and $^{266}$KMSK$^{269}$ motifs, bound to PLP (Supplementary Fig. 13). The sequence data showed that several Lys residues, especially at the KIIK and KMSK motifs, are conserved in EcCARS and other homologues from different organisms, including mammals (Fig. 2b and Supplementary Fig. 14). Also, conserved two cysteine residues bound to the active center Zn$^{2+}$ (Fig. 2b and Supplementary Fig. 14). To clarify the function of PLP bound to EcCARS, we constructed a series of Lys mutants of this enzyme (Supplementary Table 4) and measured enzyme activities in terms of persulfide, i.e., CysS–(S)$_n$–H, formation and protein synthesis or translation. We observed, via the HPE-IAM labeling LC-MS/MS analysis, a marked decrease in CysSSH and CysSSSH synthesis, compared with the wild type (WT), for various Lys to Ala mutants at K73A, K76A, K266A, K269A, and double mutants K73/76A and K266/269A of EcCARS (Fig. 2c), all of which had intact protein synthesis

potential as assessed by the PUREfrex cell-free protein synthesis assay (Fig. 2d). We also quantified the amounts of PLP bound to EcCARS by LC-ESI-MS/MS using 2,4-dinitrophenylhydrazine (DNPH). The DNPH-labeling LC-MS/MS analysis indicated that the amounts of PLP bound to WT EcCARS and four different Lys mutants correlated well with their CPERS (persulfide producing) activities (Supplementary Fig. 13b). In contrast, cysteine to aspartate mutants such as C28D (also C28S) and the double C28/209D mutant still maintained high persulfide production, similar to that of the WT cells (Fig. 2e), albeit their protein synthesis and translational activity were strongly attenuated (Fig. 2f).

Our computational modeling of the three-dimensional structure of EcCARS supported PLP binding to the particular Lys residues at the $^{73}$KIIK$^{76}$ and $^{266}$KMSK$^{269}$ motifs of EcCARS (Fig. 3a). The present computational simulation predicts two potential PLP-binding sites at K73 and K269 of KIIK and KMSK motifs. Also, this modeling revealed that PLP-bound motifs have a vicinal location within 10–20 Å distance but apparently distinct from both the ATP-binding HIGH motif and the Zn$^{2+}$-binding active site of the EcCARS for Cys-tRNA$^{Cys}$ biosynthesis. A commensurate change in the binding capacity and/or stability of

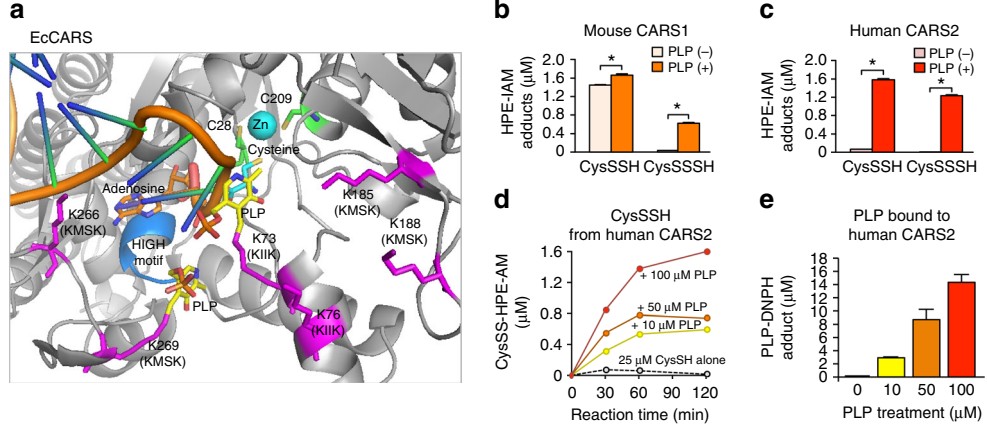

**Fig. 3** Computational modeling of EcCARS structure, and CysS–(S)$_n$–H biosynthesis by CARS1/2. **a** A molecular docking model of PLP-bound EcCARS generated by SwissDock using the crystal structure of EcCARS (PDB ID: 1LI5). Cysteinyl-tRNA is placed by superimposing the crystal structure of the EcCARS-Cysteinyl-tRNA binary complex (PDB ID: 1U0B) to the docking model. **b**, **c** PLP-dependent CysSSH and CysSSSH biosynthesis by mouse CARS1 and human CARS2. CysSSH and CysSSSH production was quantified by means of HPE-IAM labeling LC-MS/MS analysis in the reaction of recombinant mouse CARS1 and human CARS2 (200 µg/ml each) with 25 µM L-cysteine in the presence or absence of 100 µM PLP (37 °C, 2 h). The data are means ± s.d. ($n = 3$). *$P < 0.01$. **d** Concentration-dependent effects of PLP on CysSSH and CysSSSH production by recombinant human CARS2. Human CARS2 (200 µg/ml) reacted with 25 µM cysteine in the presence of 0, 10, 50, or 100 µM PLP at 37 °C for 30–120 min. No appreciable cysteine persulfide production was detected in the reaction mixture of cysteine and PLP alone as long as no >100 µM PLP was used. **e** Precisely quantitative analysis for PLP bound to human CRAS2. Human CARS2 treated with various concentrations of PLP (**d**) at 37 °C for 1 h was reacted with DNPH to form PLP-DNPH adduct, followed by quantification by LC-ESI-MS/MS analysis

PLP seems to exist, caused by the mutation of any one of four Lys residue among four Lys residues because each single Lys mutation at the KIIK and KMSK motifs greatly affected all CysS–(S)$_n$–H synthesis activity of EcCARS (Fig. 2c). One possible explanation for the commensurate effect is that PLP may need multiple Lys residues, rather than a single Lys binding, to exhibit stable binding and full catalytic activity of CARS to function as CPERS during CysS–(S)$_n$–H formation. That is, for their stable binding and catalytic activity, PLP-dependent catalytic activity may need stabilization by a multiple Lys binding, because CysSSH produced by CARS, due to its highly nucleophilic nature, may readily interfere with the electrophilic aldehyde group of PLP to form an imine (Schiff base) linkage on the Lys residues, which would cause instability of the catalytic activity of PLP bound to these particular Lys residues of CARS. This interpretation receives support from by the aforementioned computational structural analysis showing the close localization (in 20 Å) of these Lys residues at KIIK and KMSK motifs (Fig. 3a). Together these data suggest that EcCARS is indeed an efficient CPERS enzyme with independent catalytic functions in aminoacyl-tRNA biosynthesis.

**CARS2 functions as a CPERS conserved in mammals**. Two different CARSs exist in mammals: CARS1 (cytosolic) and CARS2 (mitochondrial)[30–32]. Both CARSs (mouse CARS1 and human CARS2, which we tested herein) had strong CysS–(S)$_n$–H producing activities, which depended on the presence of PLP (Fig. 3b–d). Also, a very nice correlation was found between the CPERS activity and PLP content of CARS2 containing varied amounts of PLP incorporated after treatment with different concentrations of PLP (Fig. 3e). To clarify how much cellular CysS–(S)$_n$–H originated from CARS1 and CARS2 in human cells, we attempted to disrupt CARS1 and CARS2 genes in HEK293T cells via the CRISPR/Cas9 system in HEK293T cells. We could not obtain CARS1-knockout (KO) cells, but we successfully established CARS2 KO cells. We selected one of the clones, carrying a 30-bp deletion plus an 8-bp insertion just downstream of the translation-initiating codon in the CARS2 first

exon, was selected for LC-MS/MS analysis (Supplementary Fig. 15). CysS–(S)$_n$–H and GSSH levels decreased significantly in CARS2 KO cells (Fig. 4a, b), which suggests that CARS2 is a major producer of persulfide. Because we still detected a low level of CARS2 in CARS2 KO cells (Fig. 4c), we also treated the cells with siRNA against CARS2, which resulted in the 67 and 42% decreases in CysSSH and GSSH levels, respectively (Fig. 4a, b). When we knocked down CARS1 in CARS2 KO cells, CysSSH decreased only marginally, which suggests a predominant role of CARS2 in the production of CysSSH. Immunoblot analysis and immunostaining verified the reduced CARS2 and CARS1 protein levels in CARS2 KO cells and in cells with CARS1 or CARS2 siRNA (Fig. 4c and Supplementary Figs 16 and 17).

Markedly reduced persulfide formation in CARS2 KO cells was recovered by adding back WT CARS2. CARS2 C78/257D mutant rescued the persulfide production of CARS2 KO cells, but K124/127A, and K317/320A mutants (mutants of KIIK and KMSK motifs, respectively), did not (Fig. 4d, e). The CARS2 KO cells had a markedly decreased Cys-tRNA synthetase activity, and again adding back the C78/257D mutant resulted in lost Cys-tRNA synthetase activity, as assessed by the expression of mitochondrial cytochrome c oxidase subunit 1 (MTCO1 encoded by mitochondrial DNA), but still retained full CPERS activity; conversely, K124/127A and K317/320A mutants had impaired CPERS functions but retained Cys-tRNA synthetase activity (Fig. 4f, g). These results clearly verify that CARS2 truly functions as a CPERS in mammals and that this function is separate from cysteinyl-tRNA synthetase activity.

We also evaluated the potential contribution of CSE and CBS to the endogenous persulfide production in HEK293T cells. Silencing of CSE and CBS suppressed the persulfide production, but notably, intracellular cysteine (CARS substrate) levels were significantly decreased (Supplementary Fig. 18). In CARS2 KO cells, knockdown of CSE and CBS also reduced cysteine levels but not persulfide production (Supplementary Fig. 18). Therefore, cysteine production is dependent on both CSE and CBS, and thus cysteine is provided via the metabolic pathways mediated by CSE/CBS in each cell line irrespective of CARS2 expression. In

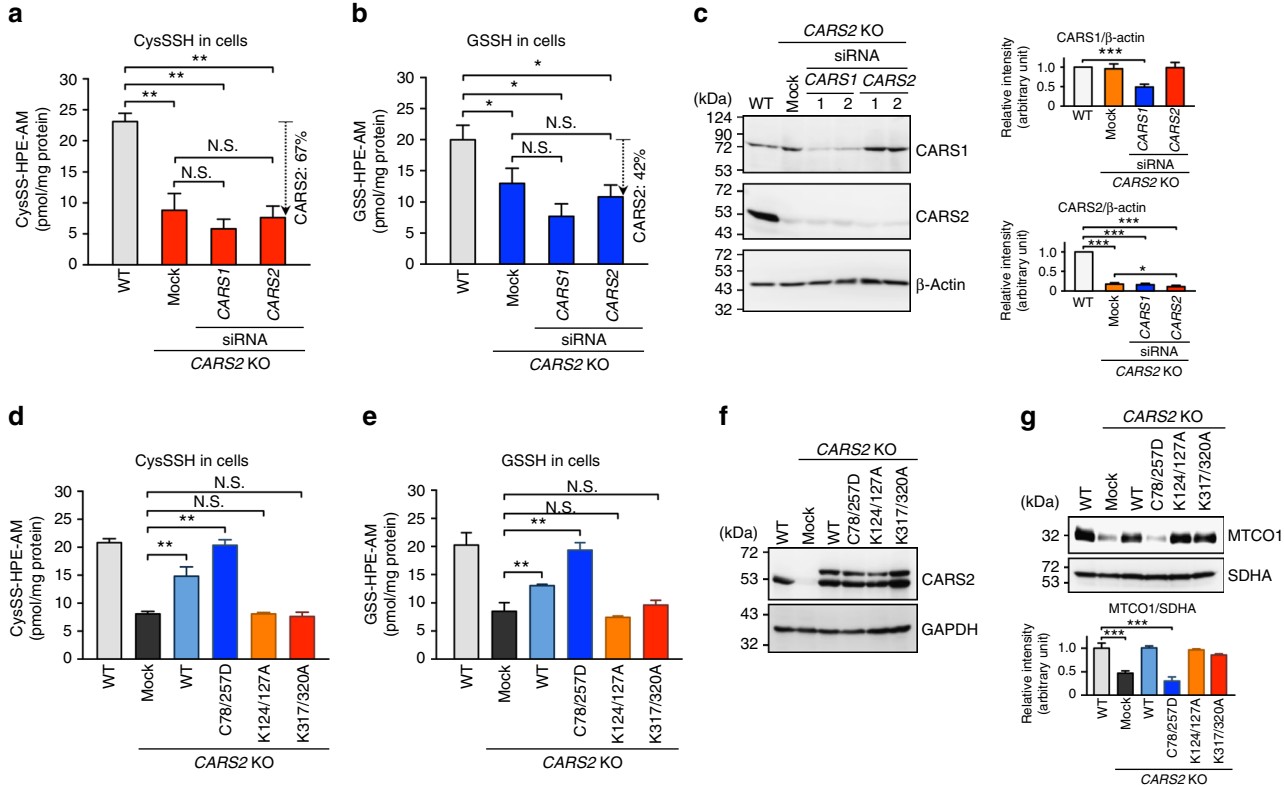

**Fig. 4** Endogenous formation of persulfides in HEK293T cells. Intracellular levels of CysSSH (**a**) and GSSH (**b**) in WT and *CARS2* KO cells with *CARS1* or *CARS2* knocked down. Data are means ± s.d. (*n* = 3). \*P < 0.05; \*\*P < 0.01; N.S., not significant. **c** CARS1 and CARS2 Western blotting for cells used in **a** and **b**. Lane 1 and 2, duplicate determinations with each siRNA. The right panel shows the densitometric analysis for the western blot shown in the right panel. The data are means ± s.d. (*n* = 3). \*\*\*P < 0.001. Production of CysSSH (**d**) and GSSH (**e**) in *CARS2* KO cells with WT or CARS2 C and K mutants added back. The data are means ± s.d. (*n* = 3). \*\*P < 0.01; N.S., not significant vs. *CARS2* KO mock. **f** CARS2 western blotting for WT and *CARS2* KO cells with WT or *CARS2* C and K mutants added back. **g** Western blotting for the cells in **d** and **e** with different mitochondrial proteins: MTCO1, mitochondrial cytochrome *c* oxidase subunit 1 (encoded by mitochondrial DNA) and SDHA, succinate dehydrogenase complex flavoprotein subunit A (encoded by genomic DNA). Supplementary Fig. 16 provides full blot images. The lower panel shows the densitometric analysis for the western blot. The data are means ± s.d. (*n* = 3). \*\*\*P < 0.001

addition, almost two thirds of CysSSH seems to be supplied by CARS2 in HEK293T cells based on the decrease by almost two thirds in the CysSSH levels. The rest of CysSSH in the *CARS2* KO cells were not derived from CSE/CBS expressed in HEK293T cells, since no further reduction of CysSSH was obtained even by CSE/CBS knockdown in *CARS2* KO cells. These results suggest that CSE and CBS do not contribute directly to persulfide production but rather may promote the biosynthesis of cysteine and its supply to CARS, at least in this cultured cell model under physiological conditions.

To further clarify CPERS functions of CARS2 in vivo, we generated the *Cars2*-deficient mice by using CRISPR/Cas9 technology. As Fig. 5 illustrates, a guide RNA (gRNA) was designed against exon 1 of *Cars2*. We established a mutant mouse line with a mutant *Cars2* allele (line 1) that had a 200-bp deletion containing a translation-initiating codon in exon 1 (Fig. 5a, b). Mating of F1 *Cars2* heterozygous KO (*Cars2*[+/−]) mice produced WT and *Cars2*[+/−] mice, but not homozygous mice (viable offsprings included 20 WT mice and 19 *Cars2*[+/−] mice), which suggests that *Cars2*[−/−] mice are embryonic lethal. *Cars2*[+/−] mice were normally born without any apparent abnormalities in macroscopic appearance or growth profiles during the observation period of at least 6 months after birth, but they demonstrated reduced mitochondrial expression of CARS2 protein by half and marked attenuation of CysSSH production; in contrast, we

observed no appreciable change in mitochondrial DNA-encoded MTCO1, which indicated intact Cys-tRNA synthetase activity in *Cars2*[+/−] mice (Fig. 5c–e and Supplementary Fig. 19a). Therefore, we quantified the sulfide metabolites in the liver of *Cars2*[+/−] mice and their WT littermates via LC-MS/MS analysis with HPE-IAM as described earlier. As we expected, *CARS2*[+/−] mice showed a striking difference in persulfide production compared with the WT littermates (Fig. 6a, b). Endogenous levels of CysSSH and all other derivatives (e.g., GSSH, HS⁻, thiosulfate, and hydropolysulfides) decreased by 50% or more in the liver and lung of *Cars2*[+/−] mouse compared with WT mice.

To exclude the possibility of off-target effects by the gRNA used to produce line 1 *Cars2*[+/−] mice, we developed another strain of *Cars2*[+/−] mice (line 2) with an alternative gRNA targeting *Cars2* exon 3. Line 2 *Cars2*[+/−] mice had phenotypes almost identical to those of line 1 (Supplementary Figs. 20 and 21).

That heterozygous *Cars2* mutant mice manifested a CysSSH reduction by ~50% should be noted; it suggests that *Cars2* contributes almost entirely to the CysSSH production in mouse tissues under physiological conditions. As an important finding, *Cars2* disruption did not alter expression levels of other sulfide-metabolizing enzymes, including CSE, CBS, and 3-mercaptopyruvate sulfur transferase (3-MST) (Fig. 5e, Supplementary Figs. 19b, c and 21), which emphasized the sole

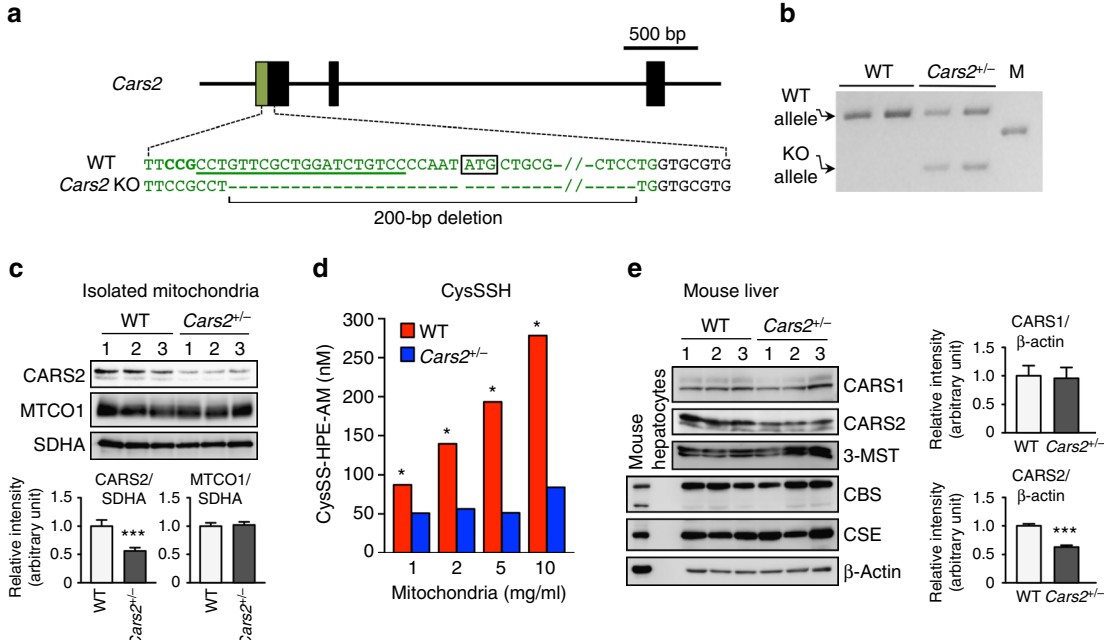

**Fig. 5** Generation of *Cars2*-deficient mice via the CRISPR/CAS9 system. **a** Schematic illustration of the mouse *Cars2* gene structure and sequences of WT and mutant alleles around the target locus. Green and black letters indicate the first exon and intron of *Cars2*, respectively. The targeted locus of gRNA and protospacer-adjacent motif (PAM) sequence were indicated in the WT sequence are indicated by underlined and bold letters, respectively. A modified allele sequence obtained from the *Cars2*-edited mouse (line 1) is shown below. **b** Detection of mutations introduced by gRNA-Cas9 targeting *Cars2* via PCR with genomic DNA from WT and *Cars2+/−* mice. *Cars2+/−*, *Cars2* heterozygous KO mice, M: DNA molecular weight marker. **c** Western blotting of CARS2 and mitochondrial proteins, e.g., MTCO1 and SDHA, from mitochondria isolated from the liver. The lower panel shows the densitometric analysis of the western blot. Data are means ± s.d. ($n = 3$). ***$P < 0.001$. **d** CysSSH production in mitochondria isolated from the liver of WT and *Cars2+/−* littermate mice. Various concentrations of isolated mitochondria were reacted with HPE-IAM for 1 h, followed by LC-MS/MS analysis (see Supplementary Methods for details). Mitochondria were obtained from line 2 *Cars2+/−* mice (Supplementary Figs. 20 and 21). *$P < 0.05$, WT vs. *Cars2+/−* mice (two-way ANOVA). **e** Western blotting of CARS1, CSE, CBS, and 3-MST with liver tissue obtained from WT and *Cars2+/−* mice. Supplementary Fig. 19 provides full blot images. The right panels show the densitometric analysis of the CARS1 and CARS2 immunoblots. Data are means ± s.d. ($n = 3$). ***$P < 0.001$

contribution of CARS2 to endogenous persulfide biosynthesis in vivo.

To explore the possibility that CARS2, a mitochondrial protein, can produce CysSSH and provide it to the whole cell, we isolated mitochondria from mouse liver and measured the release of de novo-synthesized CysSSH from the mitochondria (Supplementary Fig. 22). A large fraction of CysSSH was indeed released from mitochondria, which supports the idea that CysSSH produced in mitochondria is released into the cytoplasm and maintains protein polysulfidation. As expected, CysSSH derived from whole-cell proteins was decreased in *Cars2+/−* mice, but cysteine (CysSH) did not (Fig. 7). Specifically, formation of 20–30% of CysSSH in all cell proteins (polysulfidation) depended on CARS2 expression not only in the in vivo experiment using *Cars2* KO mice (Fig. 7a) but also in the in vitro cell culture study (Fig. 7b), as identified by HPE-IAM labeling LC-MS/MS analysis with the whole cell and tissues proteins isolated. These results suggest that CysSSH derived from CARS2 significantly contributes to the polysulfidation of the whole-cell proteins. Because protein polysulfidation appears to be mediated via post-translational as well as co-translational processes, the former being controlled by the thioredoxin (Trx)–Trx reductase (TrxR) system as recently reported[4], we expect that CysSSH generated in mitochondria is released into the cytoplasm and supplies sulfur to proteins for polysulfidation (Fig. 7c). Our current evidence is the first demonstration that unequivocally verified in human cultured cells and in vivo in mice that CARS2 is the major enzyme for persulfide biosynthesis and thus functions as a CPERS in mammals.

**CARS-mediated polysulfidation and mitochondrial physiology.** Unexpectedly, *CARS2* KO cells showed markedly altered mitochondrial morphology (i.e., shrunken or fragmented appearance), which greatly improved when CARS2 was added back, as seen with the MitoTracker Red fluorescent mitochondrial stain (Fig. 8a and Supplementary Fig. 17c), transmission electron microscopy (Fig. 8b), and immunofluorescence staining for translocase of outer mitochondrial membrane 20 (TOMM20) and CARS2 (Supplementary Fig. 17a, b). Not only WT CARS2 but also the C78/257D mutant induced a strikingly improved mitochondrial morphology, but other Lys mutants tested did not (Fig. 8a, b and Supplementary Fig. 17c). In line with these findings, deletion of *CARS2* activated dynamin-related protein (Drp1), a major mediator of mitochondrial fission[33], and Drp1 GTPase activity was significantly attenuated by adding back the WT CARS2 and C78/257D mutant, thereby producing CysSSH without CARS activity, but not by adding back the K317/320A mutant (Fig. 8c). Usually, Drp1 in HEK293T cells was extensively polysulfidated (Fig. 8d), as evidenced by our new biotin-PEG-MAL capture method (Supplementary Fig. 3b). However, Drp1 polysulfidation was markedly suppressed by both *CARS2* KO and additional *CARS1/2* double-knockdown, respectively (Fig. 8d and Supplementary Fig. 23). Because Drp1 is likely activated via chemical depolysulfidation or a post-translational process operated physiologically by the Trx–TrxR system, for example, we identified Drp1 as a major signal effector molecule reversibly regulated through a unique polysulfidation and depolysulfidation process (Fig. 8e).

We next examined CARS2 contribution to mitochondrial biogenesis and function. Mitochondrial DNA normalized against

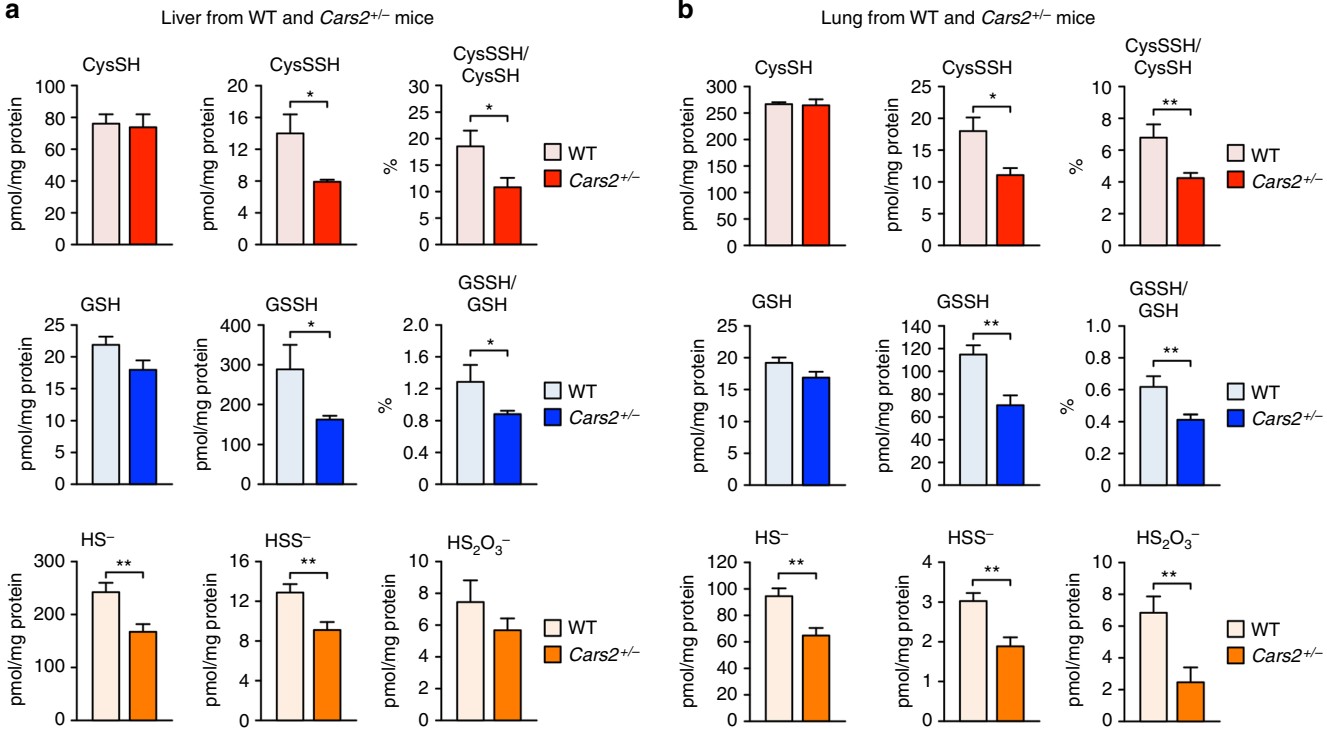

**Fig. 6** In vivo formation of various sulfide species in WT and Cars2[+/−] mice. Endogenous production of CysSSH and other related polysulfide compounds was identified by means of HPE-IAM labeling LC-MS/MS analysis in the liver **a** and lung **b** obtained from WT and Cars2[+/−] littermates (21-week-old males). The data are means ± s.d. ($n = 3$). *$P < 0.05$; **$P < 0.01$

nuclear DNA was reduced in CARS2 KO cells, which was similarly restored by WT CARS2 and C78/257D but not by Lys mutants (Supplementary Fig. 24a), which suggests that CARS2-derived persulfide enhances mitochondrial biogenesis. Mitochondrial membrane potential was decreased in CARS2 KO cells, but it recovered or even increased when the WT and C78/257D mutant were added back or overexpressed but not when Lys mutants were used (Fig. 8f and Supplementary Fig. 25a). We also used an extracellular flux analyzer to measure the oxygen consumption rate (OCR) in HEK293T CARS2 KO cells. The OCR in CARS2 KO cells was ~50% of that in WT cells (Fig. 8g), consistent with the incomplete elimination of CARS2 protein and thereby attenuated expression of MTCO1 in CARS2 KO cells (Fig. 4g). The decrease of OCR in CARS2 KO cells was recovered by introduction of WT CARS2 and C78/257D mutant but not by Lys mutants (Fig. 8g and Supplementary Fig. 25b). A novel concept emerging from these observations is that CARS2-derived cysteine persulfides play an important role in the electron transport chain (ETC) in mitochondria, which sheds light on a completely new and fundamental role of persulfides in supporting mitochondrial bioenergetic function.

**CARS2 linked up to mitochondrial ETC.** In our efforts to elucidate the mechanism of how CARS2-derived CysSSH contributes to the mitochondrial bioenergetics function, we noticed a quite different profile of the products of human CARS2 in the cell-free enzyme reaction compared with cellular CARS2 metabolism in HEK293T cells in culture (Fig. 9a, b). Although CARS2 synthesized mostly CysSSH/SSSH in a cell-free solution (Fig. 3c, d), preferential formation of HS− ($H_2S$) together with thiosulfate ($S_2O_3^{2−}$) over CysSSH was evident with HEK293T cells. We thus hypothesized that the mitochondrial compartment is a unique metabolic environment in which de novo CysSSH synthesized by

CARS2 may be further metabolized, possibly being coupled with the mitochondrial ETC.

To understand how the ETC function and CysSSH derived from CARS2 are associated (Fig. 8g and Supplementary Fig. 25b), we examined the effect of ETC suppression on the metabolic profile of CysSSH and its derivatives in HEK293T cells (Fig. 9c–h). We then used two approaches to inhibit the ETC in the cells: one method was to use a specific inhibitor of complex III, antimycin A (Fig. 9c–e), and the other ETC disrupter used was ethidium bromide to induce mitochondrial DNA deprivation (Fig. 9f–h and Supplementary Fig. 24b; see Supplementary Methods for details). Both ETC suppressive treatments caused a significant increase in CysSSH and simultaneous reduction of HS− production, as assessed by the HPE-IAM labeling LC-MS/MS analysis (Fig. 9c–h). These inverse and stoichiometric relationships between CysSSH and hydrosulfide anion (HS−) formation strongly suggested an ETC activity-dependent conversion of CysSSH to HS− mediated via the ETC occurring in the cells (Fig. 9e, h). We interpret these results to mean that CysSSH derived from CARS2 in mitochondria is effectively reduced by accepting an electron from the ETC to release HS− ($H_2S$), as Fig. 9i illustrates.

These data thus provide robust support for the idea that the CARS2-CysSSH pathway is involved in the mitochondrial function because CARS2-dependent CysSSH production is functionally integrated into and tightly linked to the mitochondrial ETC, which is in turn involved in the energy metabolism, as Fig. 10 illustrates. In fact, low (nM) concentrations of $H_2S$ reportedly sustained the ETC function possibly mediated by sulfide:quinone reductase and other potential enzymes that oxidize sulfides to thiosulfate ($S_2O_3^{2−}$)[7,34–38]. How $H_2S$ is supplied endogenously in mitochondria remained unclear, however. Our earlier and current studies suggest that CSE, CBS, and 3-MST are not major sources of $H_2S$ in mitochondria in

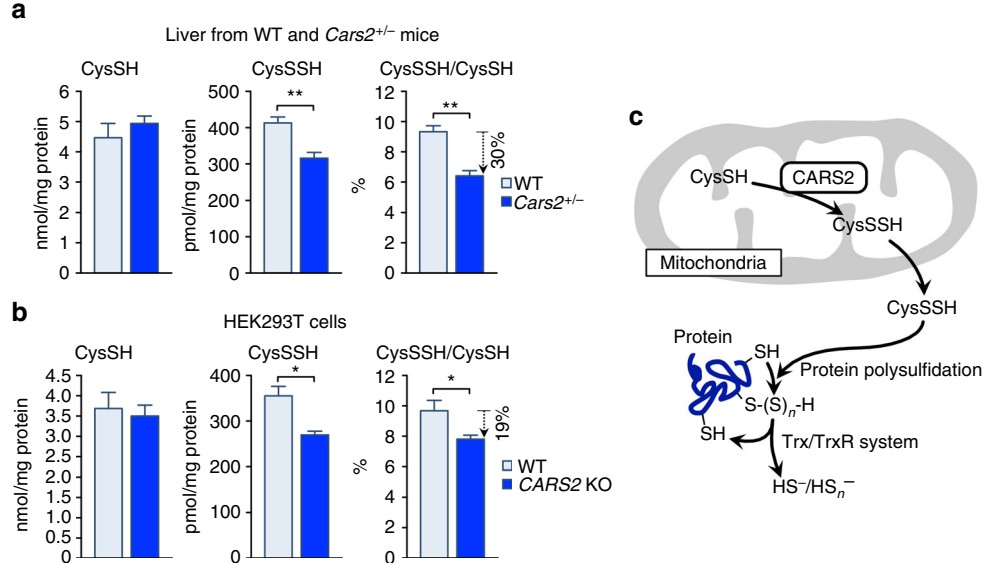

**Fig. 7** Endogenous protein polysulfidation in vivo and in HEK293T cells. The amounts of CysSSH formed in whole cell protein recovered from the mouse livers of WT and $Cars2^{+/-}$ (line 1) 21-week-old male littermates (**a**) and from WT and CARS2 KO HEK293T cells (**b**) were quantified by using HPE-IAM labeling LC-MS/MS analysis. Data are means ± s.d. ($n = 3$). *$P < 0.05$; **$P < 0.01$. **c** Schematic drawing of the mechanism of the extramitochondrial release of CysSSH into the cytosol, which may regulate whole cell protein polysulfidation

various mammalian cell lines and in mice in vivo (Fig. 5e and Supplementary Fig. 18)[7,20–24]. In this context, our study is the first to verify that HS⁻ (or H₂S) is indirectly formed from CARS2 via CysSSH generation in the mitochondrial environment (Figs. 9i and 10). Moreover, our recent study determined that CysSSH contributed to the endogenous formation of iron-sulfur clusters[14]. Because iron-sulfur clusters are known to be synthesized and utilized in complexes I-III of the ETC in mitochondria[39], and are actively transported extramitochondrially, the CysSSH-dependent HS⁻ metabolism may be coupled with the generation of iron-sulfur centers of the mitochondrial ETC and cytosolic formation and maintenance of various iron-sulfur complex machineries as well. Our reasonable conclusion is, therefore, that CARS2 functions as a major CPERS, which in turn promotes mitochondrial biogenesis and bioenergetics (Fig. 10).

## Discussion

Until now, endogenous persulfides were thought to be formed as a result of H₂S/HS⁻ oxidation via post-translational processes, and serve as protein cysteine thiol-bound intermediates of detoxification enzymes[3,7,21], and as metal ligands for iron and zinc complexes[11–15]. While CSE and CBS can catalyze CysSSH biosynthesis by using cystine as a substrate[3,4,6–10,18–21], several cells and tissues without CSE/CBS expression and CBS/CSE KO mice reportedly synthesized appreciable amounts of persulfides[3,20,22–24], but the source of the persulfides (poly-sulfides) or the sulfane sulfur reservoir has remained elusive. We here demonstrate that CARSs catalyze CysS–(S)ₙ–H formation from cysteine and co-translational protein polysulfidation. Also, CSE and CBS may still play a major role in the CysSSH pro-duction via the direct catalytic reaction using cystine as the substrate especially under pathophysiological conditions asso-ciated with oxidative and electrophilic stress, where intracellular cystine concentrations are considerably approaching the high $K_m$ value of CSE[3,7,21,40–42].

The second, even more crucial, finding is that the mitochon-drion is a key cellular compartment for the formation and action of CysSSH and CysS–(S)ₙ–H. Notably, CysSSH is mostly

generated by CARS2 localized in the mitochondria and is released extramitochondrially into the cytoplasm so that it can effectively produce CysS–(S)ₙ–H and protein polysulfidation in whole-cell compartments. The current study established that CARS2-derived CysSSH (CysS–(S)ₙ–H) indeed sustains mitochondrial biogenesis and the ETC function. While the implications of these findings await further investigation, a recent clinical study by Coughlin et al. documented an intriguing result: CARS2 mutations iden-tified in a patient were associated with ETC impairment and mitochondrial dysfunctions[31]. Although the patient's clinical symptoms resulted from loss of a canonical function of CARS2, which the neurological disorders might be caused by impairment of CPERS activity of CARS2 is plausible, and thus this impaired activity may overlap with the observed impairment of Cys-tRNA aminoacylation.

The nature of sulfane sulfur or polysulfides has continued to be a puzzle for a long time, because of a complicated polysulfide chemistry with dual electrophilic and nucleophilic characteristics. Previous reports demonstrated the ability of a trisulfide species to react with numerous electrophiles. For example, Fletcher and Robson reported that thiocystine (cystine trisulfide, CysSSSCys) readily reacted with electrophilic halogens (e.g., Br₂), which resulted in cleavage of the S–S bond[25]. A review by Parker and Kharasch also discussed numerous examples of the electrophilic cleavage of the S–S bond in disulfides by electrophilic reagents such as protons, sulfenium ions, and halogens[26]. More recently (and directly relevant to our studies), Abdolrasulnia and Wood reported that CysSSSCys reacted readily with iodoacetic acid (a well-established thiol-modifying agent) to ultimately give car-boxymethylthiocysteine (CysSS–CH₂COOH)[27], which is con-sistent with the idea that a nucleophilic sulfur atom of the polysulfide reacted with the electrophilic iodoacetic acid species and led to S–S bond cleavage. Previous examination of the reaction of electrophiles with disulfides (the simplest of all polysulfides) is entirely consistent with this idea[28]. Thus, ample precedence for the nucleophilic character of polysulfides exists, by capitalizing on such a unique property, we are now able to identify the cysteine and protein polysulfidation occurring endogenously by means of a conventional PMSA or capturing

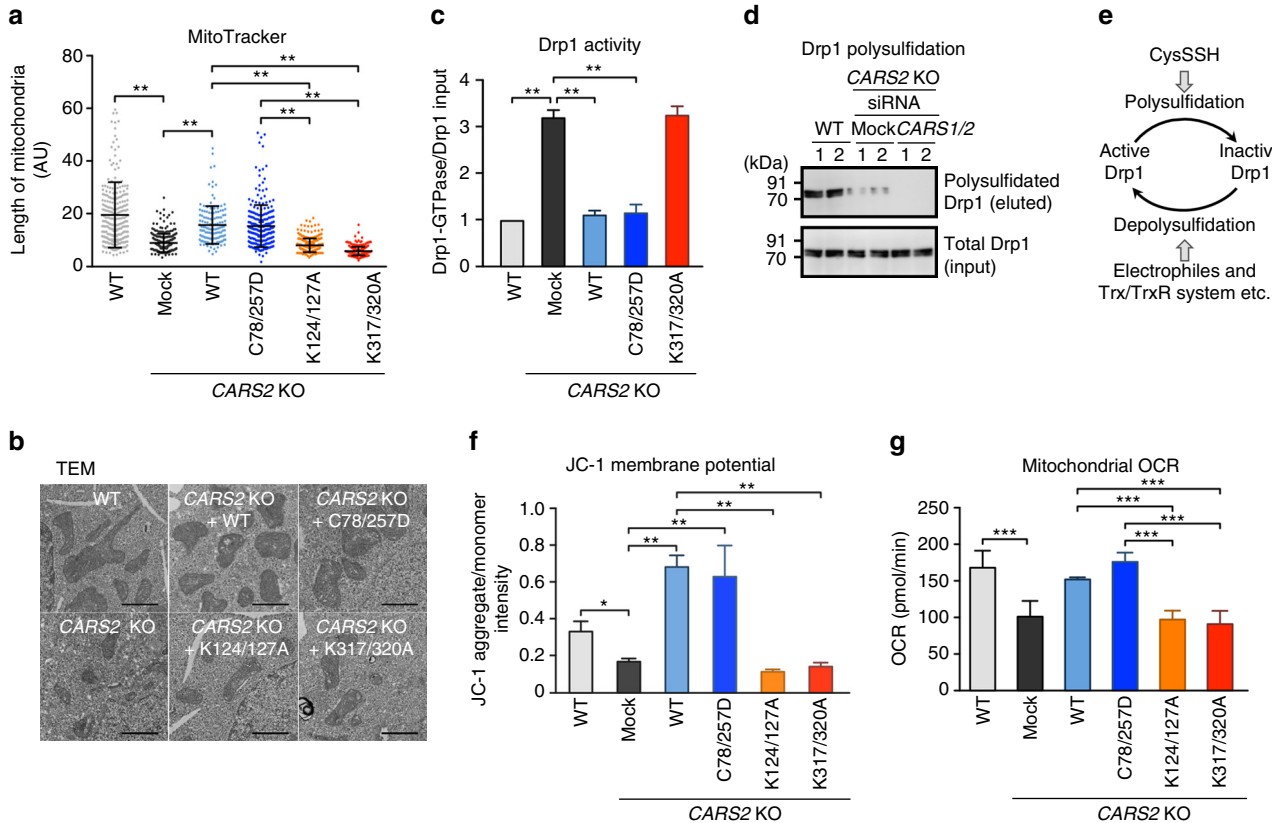

**Fig. 8** CARS2-dependent mitochondrial morphogenesis and bioenergetics. **a** Mitochondrial morphological analyses with MitoTracker Red fluorescent mitochondrial stain: morphometric analysis of mitochondrial length in HEK293T cells (WT and *CARS2* KO; *CARS2* WT and mutants added back). AU, arbitrary unit. The data are means ± s.d. (*n* = 3). **P < 0.01. **b** Transmission electron microscope (TEM) images of the cells in **a**. Scale bars, 1 μm. **c** Identification of Drp1 activity in HEK293T cells (WT and *CARS2* KO; *CARS2* WT and mutants added back). The GTP-agarose pulldown assay was performed. The data are means ± s.d. (*n* = 3). **P < 0.01. **d** Drp1 expressed in extensively polysulfidated (biotin-PEG-MAL capture method) HEK293T cells. Drp1 was markedly suppressed and nullified by *CARS2* KO and *CARS1/2* knockdown. Lanes 1 and 2 show duplicate determinations with each siRNA. Supplementary Fig. 23 provides full blot images. **e** A schematic drawing of Drp1 activity as regulated by protein polysulfidation and depolysulfidation, as affected by polysulfides vs. electrophiles and Trx/TrxR system. **f** Changes in membrane potential as assessed by using JC-1 staining of HEK293T cells (WT and *CARS2* KO; *CARS2* WT and mutants added back). The data are means ± s.d. (*n* = 3). *P < 0.05; **P < 0.01. **g** Assessment of mitochondrial electron flow in HEK293T *CARS2* KO cells with or without adding back WT and C78/257D, K124/127A, and K317/320A mutants, as analyzed by measuring OCR using an extracellular flux analyzer. Time dependence of oxygen consumption before/after inhibition of mitochondrial respiration at complexes I and III by rotenone/antimycin A, and its statistical summary; the data are means ± s.d. (*n* = 3). ***P < 0.001

assays and even by using HPE-IAM labeling LC-MS/MS analysis. The present discovery of a novel polysulfide biosynthesis, therefore, can now explain substantial endogenous generation of sulfane sulfur, which we clarified as composed of various polysulfide derivatives and which is biosynthesized by CPERSs and CARSs.

Our findings raise a number of important questions; however, for example, why are such protein-bound cysteines abundantly polysulfidated, does polysulfidation affect protein folding? And, what function does this modification play in compartments other than mitochondria? Determining how CPERS activity is regulated will also be important. Given the powerful effects of persulfides on mitochondrial morphology and bioenergetics, the availability of persulfides in cells must be subject to stringent regulation. Although CPERSs play a critical role in generating CysSSH, the Trx–TrxR system may help maintain cellular persulfide concentrations within certain limits by controlling the rate of persulfide degradation[4].

Some aminoacyl-tRNA synthetases reportedly possess functions in physiological processes besides their role in translation[43]. The mitochondria-promoting functions of CARS2 suggest its non-canonical roles and therefore may therefore represent

"moonlighting" roles of CARS2. However, CARSs effectively synthesize cysteine polysulfides, and this process is closely related to the initial translational process of de novo synthesis of nascent polypeptides in ribosomes (cf. Fig. 1b and Supplementary Fig. 10). The CPERS function of CARSs is apparently associated not only with translation but also with the mitochondrial respiration, which indicates that CARSs, rather than having a moonlighting role, have a primary function of producing persulfides.

In conclusion, our discovery of reactive persulfide production mediated by the CARS or CPERS pathway and the potent effects on mitochondrial functions observed would seem to represent a significant evolution of molecular and cell biology, thereby inviting a paradigm shift in the current understanding of cellular translation, redox signaling, and energy metabolism (Fig. 10). Our discovery of CARS and CPERS as a major sources of reactive persulfides in biology may usher in a new era of modern redox biology and life science research that hold great potential to invigorate translational studies in a variety of disease processes known to be associated with aberrant redox regulation and mitochondrial dysfunction.

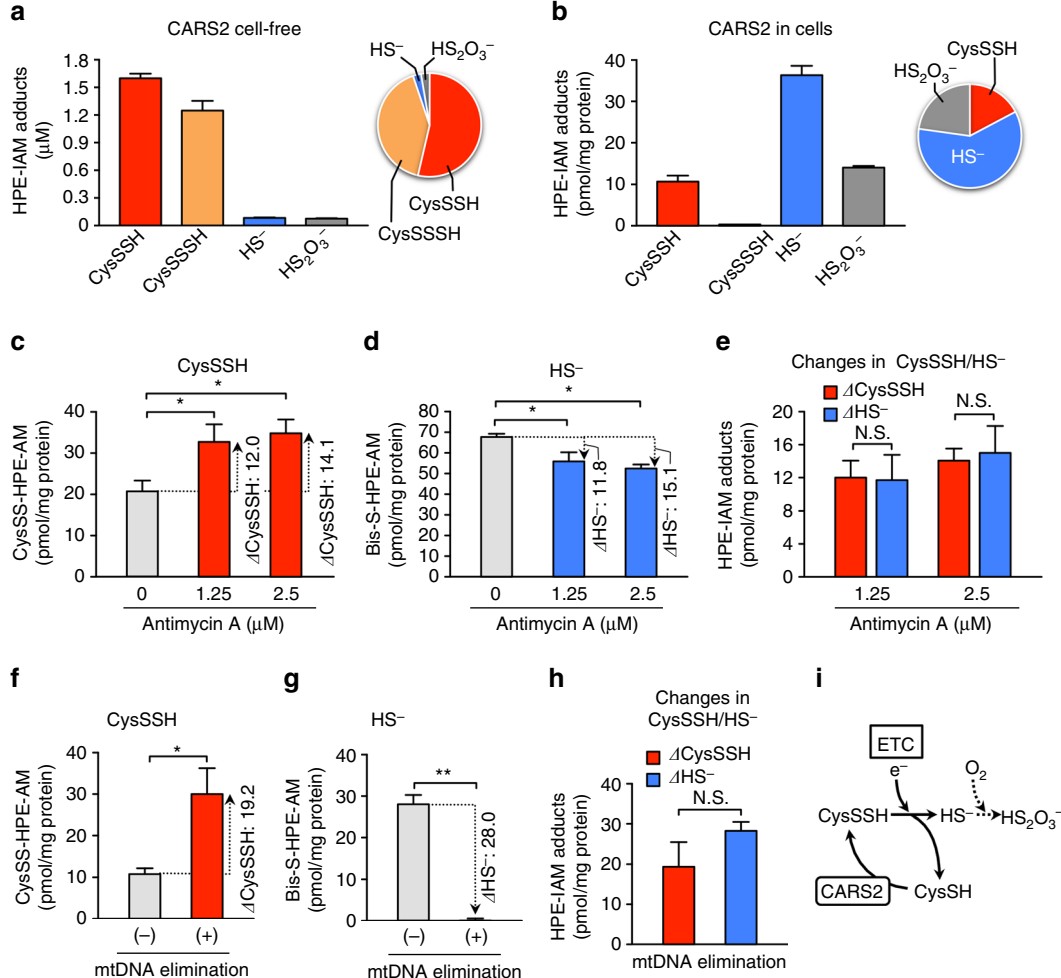

**Fig. 9** Mitochondrial ETC-mediated reduction of CysSSH. **a**, **b** Sulfide metabolite profiling for the reaction of the recombinant human CARS2 in vitro (**a**) and of CARS2 expressed in HEK293T cells (**b**). **c–h** Changes in amounts of CysSSH (ΔCysSSH) and HS$^-$ (ΔHS$^-$) induced by complex III inhibition by antimycin A **c–e** or by mitochondrial DNA (mtDNA) elimination induced by ethidium bromide (**f–h**) in WT and *CARS2* KO HEK 293 T cells. The values of CysSSH and HS$^-$ shown in **b–h** indicate the quantity of each compound produced in the cells in a manner dependent on CARS2 expression, which was determined by subtracting each amount in *CARS2* KO HEK293T cells from that in the WT cells, after quantification of each metabolite via HPE-IAM labeling LC-MS/MS analysis. **e**, **h** Stoichiometric alterations (conversion) between CysSSH and HS$^-$ in cells by the ETC inhibition. **i** Schematic diagram of ETC-mediated CysSSH reduction to form HS$^-$ and possible further conversion to $S_2O_3^{2-}$. The data are means ± s.d. ($n = 3$). *$P < 0.05$; **$P < 0.01$; N.S., not significant

## Methods

**LC-ESI-MS/MS analyses for per/polysulfides.** LC-ESI-MS/MS analysis with HPE-IAM (Supplementary Fig. 5 and Supplementary Table 2) was used to determine CysSSH or CysS–(S)$_n$-SH formed from EcCARS and CARSs. To identify CysS–(S)$_n$-H formed and incorporated into Cys-tRNA via the enzymatic reaction of EcCARS, 200 μg/ml recombinant EcCARS was reacted with 0.5 mg/ml tRNA (Sigma-Aldrich) and CysS–(S)$_n$-H or 10 μM cysteine as the substrate, in 50 mM HEPES buffer (pH 7.5) containing 1 mM ATP, 25 mM KCl, and 15 mM MgCl$_2$ at 37 °C, followed by alkylation with 1 mM HPE-IAM for 20 min at 37 °C. CysS–(S)$_n$-H were formed from 10 μM cystine and 30 μM Na$_2$S$_2$ in 30 mM HEPES buffer pH 7.5 at 37 °C for 5 min. The Cys-tRNA$^{Cys–(S)n-H}$ synthesized by EcCARS was precipitated by adding 10% trichloroacetic acid to the reaction mixture, followed by trapping by cotton wool filters (100 μl) placed in pipette tips. The precipitated total tRNA containing Cys-tRNA$^{Cys–(S)n-H}$ was washed with 10% trichloroacetic acid (200 μl twice) and with 70% ethanol (200 μl twice) to completely remove the free cysteine and CysS–(S)$_n$-H. CysS-HPE-IAM and CysS–(S)$_n$-HPE-IAM adducts were dissociated by alkaline heat hydrolysis of the ester bond of aminoacyl moieties of the Cys-tRNA$^{Cys}$ and Cys-tRNA$^{Cys–(S)n-H}$. The hydrolysis was performed in 20 mM Tris-HCl (pH 8.0), which contained known amounts of stable isotope-labeled internal standards, at 70 °C for 15 min. The eluted solutions were acidified with formic acid and analyzed via LC-ESI-MS/MS. Also, Cys-tRNA-bound CysSSH was identified by detecting a CysSSH-adenosine adduct formed in the Cys-tRNA molecules synthesized by EcCARS from the substrate cysteine. The CysSSH-adenosine adducts in the reaction of EcCARS with cysteine and Cys-tRNA were measured by using LC-ESI-MS/MS analysis. In brief, CysSSH incorporated into

tRNA as catalyzed via EcCARS with cysteine was prepared in the same manner as that described earlier, followed by alkylation with HPE-IAM and acetylation with acetic anhydride, as described earlier[44]. After precipitation and washing of samples with ethanol, the acetylated and HPE-IAM-labeled Cys-tRNA$^{CysSSH}$ was digested to generate acetylated CysSS-HPE-AM-bound adenosine by treatment with RNase ONE (Promega, Madison, WI) at 37 °C for 1 h, after which LC-ESI-MS/MS analysis was performed. To measure CysS–(S)$_n$-H generated directly by EcCARS and CARSs, recombinant EcCARS, mouse CARS1, or human CARS2 was incubated with cysteine in 50 mM HEPES buffer (pH 7.5) containing 25 mM KCl and 15 mM MgCl$_2$ with or without 1 mM ATP at 37 °C. The mixtures were then reacted with 1 mM HPE-IAM in methanol at 37 °C for 20 min to form CysS–(S)$_n$-HPE-IAM adducts. After centrifugation, aliquots of the supernatants were diluted 10–100 times with 0.1% formic acid containing known amounts of isotope-labeled internal standards and were subjected to LC-ESI-MS/MS. To clarify the molecular mechanism of CysSSH formation, 50 μM $^{34}$S-labeled L-cysteine was reacted with 200 μg/ml EcCARS as a substrate in 50 mM HEPES buffer (pH 7.5) containing 25 mM KCl and 15 mM MgCl$_2$ at 37 °C for 15–60 min. The reaction products treated with HPE-IAM were diluted with 0.1% formic acid containing known amounts of isotope-labeled internal standards, which were then subjected to LC-ESI-MS/MS as described above. To determine kinetic parameters, WT EcCARS and C28S EcCARS were incubated with different concentrations of L-cysteine in 50 mM HEPES buffer (pH 7.5) containing 25 mM KCl and 15 mM MgCl$_2$ at 37 °C for 30 s. The reaction mixtures were treated with 1 mM HPE-IAM, followed by LC-ESI-MS/MS as described above. The data were fitted by nonlinear regression to the Michaelis–Menten equation by using GraphPad Prism software ver. 6.0 (GraphPad

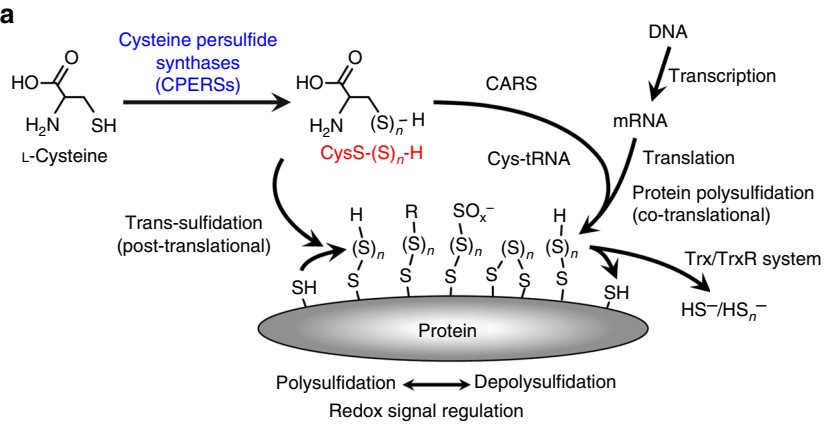

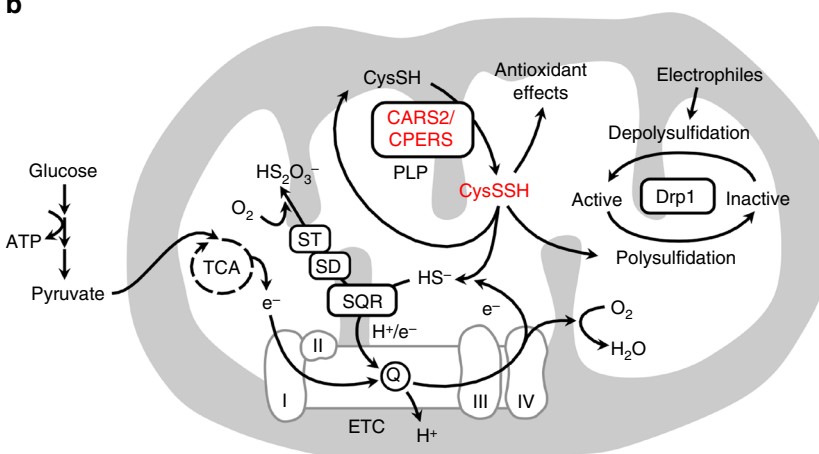

**Fig. 10** CARS-mediated protein polysulfidation and mitochondrial functions. **a** The physiological relevance of co-translational protein polysulfidation that is reversibly regulated by various post-translational modifications, including depolysulfidation. **b** A CysS–$(S)_n$–H regulation mechanism for mitochondrial functions with regard to mitochondrial biogenesis and bioenergetics. CysSSH is reductively metabolized to CysSH and HS⁻, which may be oxidized by sulfide:quinone reductase (SQR) and other enzymes, e.g., sulfur dioxygenase (SD) and sulfur transferase (ST), in a manner linked to ETC in mitochondria. The CysS–$(S)_n$–H-dependent HS⁻ metabolism may be coupled with formation of the iron-sulfur clusters, as being controlled by the mitochondrial ETC. I, II, III, and IV: complexes I, II, III, and IV; TCA tricarboxylic acid (Krebs) cycle

Software, San Diego, CA) to obtain the kinetic parameters. Each calculated enzyme parameter was compared with that of recombinant CSEs (rat and human), which we obtained from the enzymatic reaction with L-cystine as the substrate, according to our previous report[3]. For analysis of intracellular persulfide levels in cultured HEK293T cells, and livers and lungs obtained from WT and *Cars2*[+/−] littermate mice, the cultured cells and mouse tissues were lysed or homogenized in a cold methanol solution containing 1 mM HPE-IAM, after which cell lysates were incubated at 37 °C for 20 min. After centrifugation, aliquots of the supernatants of the lysates were diluted 20 times with 0.1% formic acid containing known amounts of isotope-labeled internal standards, which were then analyzed via LC-ESI-MS/MS for per/polysulfide determination. A triple quadrupole (Q) mass spectrometer LCMS-8050 (Shimadzu) coupled to the Nexera UHPLC system (Shimadzu) was used to perform LC-ESI-MS/MS. Per/polysulfide derivatives were separated by means of Nexera UHPLC with a YMC-Triart C18 column (50 × 2.0 mm inner diameter) under the following elution conditions: mobile phases A (0.1% formic acid) with a linear gradient of mobile phases B (0.1% formic acid in methanol) from 5 to 90% for 15 min at a flow rate of 0.2 ml/min at 40 °C. MS spectra were obtained with each temperature of the ESI probe, desolvation line, and heat block at 300, 250, and 400 °C, respectively; and the nebulizer, heating, and drying nitrogen gas flows were set to 3, 10, and 10 liters/min, respectively. Various per/polysulfide derivatives were identified and quantified by means of multiple reaction monitoring (MRM). Supplementary Table 2 summarizes the MRM parameters for each derivative.

**Identification of CysS–$(S)_n$–SH formed in nascent peptides**. CysS–$(S)_n$–SH species synthesized endogenously and formed in nascent polypeptides by EcCARS in *E. coli* cells in culture were analyzed by means of puromycin-associated nascent chain proteomics (PUNCH-P)[29], which was specifically modified here for

polysulfidated proteins (PUNCH for Polysulfide Proteomics, henceforth called PUNCH-PsP). The *E. coli* JM109 cells transfected with an hGAPDH expression vector (pGE-30) were cultured and hGAPDH expression was induced with IPTG as described earlier, followed by collecting and sonication of the cells in cell lysis buffer containing 0.3 mg/ml lysozyme and 2 mM IAM without any reducing agents. The supernatant obtained by centrifugation was applied to the Ni-NTA agarose column for purification of the mature GAPDH protein. From the resultant pellet of the *E. coli* cell lysate, the ribosomal fraction was isolated via sucrose density gradient ultracentrifugation, as reported previously[29]. The ribosomal fraction was suspended in polysome buffer (50 mM Tris-HCl, pH 7.5, 10 mM MgCl₂, and 25 mM KCl), containing an EDTA-free protease inhibitor cocktail (as indicated by the manufacturer), and was then reacted with 2 mM IAM at room temperature for 30 min. After the ribosomal fraction was washed with the polysome buffer, the ribosomes were treated with 5′-biotin-dC-puromycin (Jena Bioscience, Jena, Germany) in TTBS (20 mM Tris-HCl, 150 mM NaCl, 0.1% Tween 20, pH 7.6) at 37 °C for 15 min and were then reacted with avidin magnetic beads (Wako Pure Chemical Industries) to finally capture the newly synthesized poly-peptides in ribosomes in the *E. coli* cells in culture. The puromycin-labeling con-ditions were optimized for the *E. coli* ribosomes used in the present study, according to the original report[29]. The CysS–$(S)_n$–H residues in GAPDH were detected by means of LC-Q-TOF-MS as described earlier, with tryptic digests of the mature GAPDH purified simultaneously and the same digest of the nascent GAPDH polypeptides within the cultured *E. coli* ribosomes captured with and recovered from the biotin-puromycin-bound avidin beads. CysS–$(S)_n$–H in the nascent polypeptides can be selectively identified by using PUNCH-PsP, which we successfully developed and describe here (Fig. 1c and Supplementary Fig. 10). During this PUNCH-PsP analysis, the cysteine and CysS–$(S)_n$–H residues located in the polysulfide exit tunnel in the ribosomes are not accessible to exogenously added IAM and can thus be protected from alkylation by IAM because of the

unique physicochemical properties of the interior structure of the polypeptide exit tunnel in the ribosome[45–47], which allowed us to obtain specific and selective identification of the intact forms of CysS–(S)$_n$–H residues in the nascent peptides present only within the ribosomes, as Supplementary Fig. 10a shows. As soon as the mature GAPDH isolated from *E. coli.* with the Ni-NTA agarose was treated by quick digestion with 10 µg/ml trypsin at 37 °C for 30 min, which was promptly subjected to the LC-ESI-Q-TOF analysis, in a similar manner as shown for the PUNCH-PsP method.

**Preparation and purification of recombinant CARS proteins**. To generate recombinant CARSs, open-reading frames of these genes were transferred into AG1 (Agilent Technologies, Santa Clara, CA) competent cells. Recombinant EcCARS, mouse CARS1, and human CARS2 proteins were purified by using the following standard procedure. Briefly, these proteins were produced in AG1, and they were purified by using nickel nitrilotriacetic acid agarose; resultant purified proteins were extensively dialyzed against phosphate buffer and stored at −80 °C until use. Protein concentration was determined by using the Protein Assay CBB Solution (Nacalai Tesque, Kyoto, Japan), and protein purity was confirmed via SDS-PAGE.

**Generation of *CARS2* KO cell lines**. The genome editing CRISPR/Cas9 system was used to generate human *CARS2* KO cell lines. To obtain gRNA, which is highly specific for the first exon of the human *CARS2* locus and has fewer off-target sites within the human genome, we based an optimal gRNA design on the software program CRISPRdirect[48]. To express Cas9 and gRNA in HEK293T cells, the pX459 V2.0-CARS2 gRNA vector was created by inserting annealed oligonucleotide pairs (5′-caccTGGGCCTTGGGCGGGCTGGG-3′ and 5′-aaacCCC AGCCCGCC-CAAGGCCCA-3′) into the BpiI sites of pX459 V2.0. pX459 V2.0 vector, which enables expression of a gRNA (directed to the *CARS2* exon 1; Supplementary Fig. 15), SpCas9, and a puromycin resistance gene from a single vector, was obtained from the Zhang laboratory via Addgene plasmid 62988[49]. HEK293T cells were plated in 6-well plates ($1.0 \times 10^5$ cells per well) 24 h before transfection. Cultured cells were transfected with 2 µg of pX459 V2.0-CARS2 gRNA by using Lipofectamine 2000 (Invitrogen, Carlsbad, CA). The medium was changed 24 h after transfection. After another 24 h of incubation, the cells were replated on 10-cm dishes and cultured for various time periods at 37 °C with a selection medium containing 2.0 µg/ml puromycin (Invitrogen). Puromycin-resistant clones were arbitrarily selected and used for screening *CARS2* KO cell lines to finally obtain stable *CARS2* KO cell lines. Disruption of the *CARS2* gene was verified by loss of CARS2 protein expression as determined by western blotting.

**Construction of mammalian *hCARS2* expression vectors**. To produce an *hCARS2* expression vector (pPyCAGIP-FLAG-hCARS2), the XhoI fragment of pET-15b-hCARS2 was cloned into the XhoI site of pPyCAGIP-FLAG. The same vectors containing various mutant *hCARS2* genes were obtained via site-directed mutagenesis by using inverse PCR with pPyCAGIP-FLAG-hCARS2 as a template and primer sets for generation of pPyCAGIP-FLAG-hCARS2 C78/257D, K124/127A, and K317/320A.

**Transfection of various *CARS2* genes and knockdown of *CARS1/2***. WT and various mutant *CARS2* genes were transfected into HEK293T WT and mutant cells as reported recently[3] by using expression plasmids such as pPyCAGIP-FLAG-hCARS2 and CARS2 mutant vectors. Transfection of the expression plasmid was performed by using Lipofectamine 2000 according to the manufacturer's instructions. In brief, we incubated WT and *CARS2* KO HEK293T cells seeded in 24-well plates ($6 \times 10^5$ cells per well) and 8-well culture slides ($2 \times 10^5$ cells per well) for 12 h at 37 °C. For transfection, we mixed 1.5 µg per well of the expression plasmid with 50 µl of Opti-MEM (Invitrogen) in a tube. Before plasmid DNA and transfection reagent solutions were added to the cells, solutions were mixed together and incubated for 5 min at room temperature and then added to the cells, after which incubation proceeded for 30 h or 3 days. Also, knockdown of *CARS1* and *CARS2* was performed as reported recently[3] by using the following small interfering RNAs (siRNAs): CARS1, CARSHSS101368 (Invitrogen), and CARS2, CARS2HSS128464 (Invitrogen). siRNA transfection was performed by using Lipofectamine RNAiMAX (Invitrogen) according to the manufacturer's instructions. The siRNA was introduced into WT and *CARS2* KO cells, as described above for *CARS2* gene transfection.

**Generation of *Cars2*-deficient mice**. All experimental procedures conformed to "Regulations for Animal Experiments And Related Activities at Tohoku University", and were reviewed by the Institutional Laboratory Animal Care and Use Committee of Tohoku University, and finally approved by the President of University. We generated two lines of *Cars2*-deficient mice as follows. *Cars2* gRNAs vectors were constructed with use of a pT7-sgRNA and pT7-hCas9 plasmid (a gift from Dr. M. Ikawa, Osaka University)[50]. After digestion of pT7-hCas9 plasmid with EcoRI, *hCas9* mRNA was synthesized by using an in vitro RNA transcription kit (mMESSAGE mMACHINE T7 Ultra kit; Ambion, Austin, TX), according to the manufacturer's instructions. A pair of oligonucleotides targeting *Cars2* was annealed and inserted into the BbsI site of the pT7-sgRNA vector. The sequences of the gRNAs were designed as follows: 5′-GGACAGATCCAGCGAACAGG-3′ and

5′-AATAATCAAGAGAGCTAACG-3′, located at exons 1 and 3 of *Cars2* gene, to generate *CARS2*-deficient lines 1 and 2 mice, respectively. After digestion of pT7-sgRNA with XbaI, gRNAs were synthesized by using the MEGAshortscript kit (Ambion). We used C57BL/6N female mice (purchased from Crea-Japan Inc., Tokyo, Japan) to obtain C57BL/6N eggs, and we performed in vitro fertilization with these eggs. In brief, *Cas9* mRNA and gRNA were introduced into fertilized eggs by injecting using a Leica Micromanipulator System, according to the protocols reported previously[50], after which we transferred the eggs to the oviducts of pseudo-pregnant females on the day of the vaginal plug. A founder mouse harboring the *Cars2* mutant alleles was crossed with WT mice to obtain *Cars2* heterozygous mice. After segregating the *Cars2* mutant alleles, heterozygous mice with a 200-bp deletion in exon 1 (line 1) and with a 1-bp insertion in exon 3 were selected for additional analyses (Figs. 5–7; Supplementary Figs. 20 and 21).

**MitoTracker Red staining for mitochondrial morphology**. To analyze mitochondrial morphogenesis under several experimental conditions in cells, mitochondria were imaged by using the fluorescent probe MitoTracker Red CM-H$_2$Xros (Invitrogen). In brief, culture slides were coated with 0.5% polyethylene imine for more than 1 h and washed twice with PBS. *CARS2* KO cells were transfected with expression plasmids for WT and individual mutants of human CARS2 via Lipofectamine 2000. At 3 days after transfection, cultured cells were washed with Hank's buffer, incubated with 1 µM MitoTracker Red CM-H$_2$Xros at 37 °C for 30 min, rinsed twice with Hank's buffer, and examined with a Nikon EZ-C1 confocal laser microscope (Tokyo, Japan). We used ImageJ and Prism software for image processing and quantification of mitochondrial dimensions including their length.

**Mitochondrial bioenergetic functions**. To determine the membrane potential (ΔΨm) of mitochondria under several experimental conditions, tetraethylbenzimidazolyl carbocyanine iodide (JC-1) staining was performed according to the manufacturer's protocol. Accumulation of the cell-permeable JC-1 probe (Abcam) in mitochondria depends on the membrane potential, associated with a fluorescence emission shift from green to red. Briefly, WT and *CARS2* KO HEK293T cells, cultured in 8-well multichamber Millicell slides coated with PEI, were treated with various *CARS2* vectors or were untreated, as described above. For JC-1 staining, cultured cells were washed with HKRB buffer (20 mM HEPES, 103 mM NaCl, 4.77 mM KCl, 0.5 mM CaCl$_2$, 1.2 mM MgCl$_2$, 1.2 mM KH$_2$PO$_4$, 25 mM NaHCO$_3$ and 15 mM glucose, pH 7.3), incubated with 20 µM JC-1 at 37 °C for 30 min, rinsed twice with HKRB buffer, and examined with a Nikon EZ-C1 confocal laser microscope. ImageJ software was used for image processing and quantification of the JC-1 fluorescent responses.

**Mitochondrial bioenergetic functions**. Mitochondrial function was investigated, according to a previous report with a slight modification[51], by measuring the basal OCR of the mitochondria under various experimental conditions in WT and *CARS2* KO cells, using the XF96 Extracellular Flux Analyzer (Seahorse Bioscience, Agilent). At the end of the experiment, rotenone and antimycin A (2.4 µM each) were added to inhibit complexes I and III of the mitochondrial electron transport chain, respectively, to determine the remaining mitochondria-independent OCR. Net OCR was normalized to the cell number determined at the end of the experiments by means of sulforhodamine B staining (Sigma-Aldrich, St. Louis, MO). To obtain the mitochondria-specific OCR, only the rotenone/antimycin-sensitive part of cell respiration was used.

**Effect of suppression of ETC on metabolic profiles of CysSSH**. The mitochondrial ETC in HEK293T cells was inhibited either by a complex III inhibitor, antimycin A, or by elimination of mitochondrial DNA (mtDNA) induced by ethidium bromide. For the direct but partial ETC (complex III) inhibition, WT and *CARS2* KO cells were treated with various concentrations of antimycin A for 1 h, followed by methanol extraction for measurement of CysSSH and its related sulfide derivatives by HPE-IAM labeling LC-ESI-MS/MS analysis as described earlier. To indirectly suppress all ETC components (complexes), mtDNA from WT and *CARS2* KO HEK293T cells was eliminated specifically by treatment with ethidium bromide (50 ng/ml, 127 nM) for 12 days under standard cell culture conditions (37 °C, humidified, 5% CO$_2$/95% air) with DMEM containing 10% FBS, 1% penicillin-streptomycin, sodium pyruvate (1 mM), nonessential amino acids (1%), and uridine (50 µg/ml), according to a previous method with a slight modification[52]. The cells without mtDNA were then subjected to HPE-IAM labeling LC-MS/MS analysis for persulfide metabolic profiling, similar to antimycin-treated cells. The efficacy of the present mtDNA elimination and the resultant ETC suppression were assessed by measuring mtDNA as described below (Supplementary Fig. 24b), and these results were confirmed by substantial suppression of mitochondrial cytochrome *c* oxidase subunit 1 (MTCO1: encoded by mtDNA), as identified by western blotting. In contrast, MitoTracker Red staining showed no appreciable altered morphology of mitochondria in HEK293T cells with or without ethidium bromide treatment, at least under the present experimental conditions. The quantity of each sulfide produced from CARS2 in the cells was determined by subtracting the amount of each sulfide in *CARS2* KO HEK293T cells from that in the WT cells, after quantification of each metabolite via HPE-IAM labeling LC-MS/

MS analysis. Changes in the amounts of CysSSH ($\Delta$CysSSH) and HS⁻ ($\Delta$HS⁻) induced by complex III inhibition by antimycin A or by mtDNA elimination in WT and *CARS2* KO HEK293T cells were then calculated.

**Statistical analysis**. Results are presented as means $\pm$ s.d. of at least three independent experiments unless otherwise specified. For statistical comparisons, we utilized two-tailed Student's *t* test or two-way analysis of variance followed by the Student–Newman–Keuls test, with significance set at $P < 0.05$.

**Data availability**. The data that support the findings of this study are available from the corresponding author upon request.

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

## Acknowledgements

We thank J.B. Gandy for her editing of the manuscript and evaluating concepts and terminology of the paper with regard to the understanding by non-specialist readers. Thanks are also due to S. Akashi, H. Abdul Hamid, Y. Kakihana, and M. Matsubara for their technical assistance in our study. P.N. is a János Bolyai Research Scholar of the Hungarian Academy of Sciences. M.F. is supported by the National Institute for Health Research through the NIHR Southampton Biomedical Research Centre. This work was supported in part by Grants-in-Aid for Scientific Research ((S), (A), (B), Challenging Exploratory Research, and Innovative Areas "Oxygen Biology: a new criterion for integrated understanding of life") from the Ministry of Education, Sciences, Sports, and

Technology (MEXT), Japan, to T.A. (26111008, 26111001, 15K21759, 25253020, 16K15208), Y.K. (25220103), H.M. (15H04692), K.I. (26116005), T.S. (15H03115), and M.N. (15K14959); a grant from the JST PRESTO program to T.S. (10104025) and M.N. (13417243); grants from The Hungarian National Science Foundation (K 109843) and from the National Institutes of Health (R21AG055022-01) to P.N.; and grants from the National Institutes of Health (HL106598) and the National Science Foundation (CHE-1148641) to J.M.F.

## Author contributions

T.A. and H.M., experiment design, biochemistry and cell biology, data analysis, animal studies, and writing the paper; T.I., M.M.A., H.I., T.S., M.J. and T.M., MS analysis, biochemistry and molecular biology, cell biology, and cell imaging; S.W., K.I., protein structural analysis; F.-Y.W., S.K., Akir.N., H.S., N.T. and K.T., cell biology and mitochondria study; M.M. and M.O., CRISPR/Cas9 technology, cell biology, and animal studies; M.N., Akiy.N., S.F., K.Y. and Y.W., cell signaling, cell biology, and data analysis; P.N., M.F., J.M.F., chemical analysis and editing the paper.

## Additional information

**Competing interests:** The authors declare no competing financial interests.

