## [Peer Review File · Nature Communications]

PEER REVIEW FILE

Reviewers' comments:

Reviewer #1 (Remarks to the Author):

This paper addresses the biological occurrence, biosynthesis, and potential function of cysteine hydropersulfides in biological systems. Previous reports by these authors have suggested the importance of compounds containing reactive persulfides (-SSH) as potential mediators of antioxidant effects. Currently, details are lacking regarding the enzymes principally responsible for their synthesis, and how these effects are mediated. One school of thought holds that the enzymes cystathionase (CSE) and cystathionine beta synthase (CBS) readily convert cysteine to cysteine hydropersulfide. However, these authors argue that the kinetic parameters of these reactions may not be sufficient to normal cellular physiology, and that -SSH is likely being produced by other sources. Other biosynthetic routes of -CSSH formation remain to be identified.

This manuscript contains a huge amount of information, and not all findings can be summarized here. Briefly, the central finding of this manuscript is that cysteinyl tRNA synthetase may be a major source of persulfidated compounds in cells, and this activity may be particularly important to support mitochondrial function.

Among the key observations are the following:

- The authors report the development of special reagents and approaches (biotin-PEG-MAL) to detect persulfidated proteins. These reagents are employed to detect persulfidation in both over-expressed recombinant proteins, and in endogenous proteins in various cell lines. They conclude that persulfidation is common in both prokaryotes and eukaryotes.
- Next, the detection chemistry is used to show that (a) CysRS catalyzes the production of CySSH -tRNA, and (b) these aminoacylated tRNAs can incorporate persulfidated amino acids into proteins via into de novo protein synthesis.

- *E. coli* CysRS is a cysteine persulfide synthase, as are numerous other aminoacyl-tRNA synthetases. This raises the question of whether the canonical ARS amino acid binding/ active site is the locus of CysSSH activity. The authors then introduced a series of mutants at predicted conserved lysines: found that some of these important.
- The authors then address the role of the two CysRS versions in the mammalian cells: CARS1 and CARS2. Key result is that CARS activity is stimulated by PLP. Cellular levels of persulfides were reduced by CARS1 and CARS2 knockdown; these effects were rescued by the C78/257 →D mutants but not by mutants in important lysines.
- The final set of results address the interrelation of CARS persulfidation activity and overall mitochondrial metabolism. A large set of experiments shows that when CARS persulfidation activity is compromised, mitochondria function is diminished in parallel. The Add back of CARS2 of C78/257 →D but not lysine mutants led to recovery of functions. Interestingly, this mitochondrial protective function may be linked to the persulfidation of Drp1, which is known to control mitochondrial fusion and fragmentation events. They further go on to suggest that the CysSSH produced has a specific function in the electron transport chain. When mitochondria ETC is blocked by EtBR , conversion of CysSSH to HS⁻ is blocked. Suggestion is that electrons leaking out of ETC are captured to reduce CysSSH to produce HS⁻ that has protective effects,

Overall, this manuscript opens up a novel and potentially highly significant area of biochemistry whereby a standard enzyme of translation produces a new translational modification of proteins, with many implications for regulation of mitochondrial function. The wealth of information provided in this paper provides support for many of the principal conclusions of the paper. The provocative nature of this study does raise some questions and concerns that should be addressed.

Questions and Concerns:

A somewhat broad question concerns the relative importance of the two different pathways for generation of persulfidated cysteines in the client proteins. Two potential mechanisms are presented: (A) incorporation into proteins via aminoacylation and standard protein synthesis, as suggested by the data from Figure 1c,d,e and (B) modification of target proteins via a PLP dependent mechanism, as suggested by the data in Figure 2 and supp Fig 4. Depending on which of the two mechanisms predominates, there are a number of important predictions and issues.

If mechanism A predominates, this raises the question of a balance between normal cysteine and –SSH incorporation is achieved, and how wholesale incorporation of –SSH in all positions and in all proteins is avoided/regulated. Based on the known amino acid specificity of ARSs, it is

difficult to imagine other ARSs being able to carry out this function. This point is not really highlighted. Other minor question regarding this mechanism are: (i) did the authors demonstrate that this aminoacylation reaction is ATP dependent; and (ii) how does their experimental aminoacylation procedure (Supplementary page 12 an 41) rigorously limit the detection of derivatized free amino acid?

If mechanism B predominates, there are other questions that are unresolved. First, there is the claim CysRS is much more efficient producer of –CysSSH than either CSE or CB. However data in Supp Fig13 and Supp Table 2 indicate that the k_{cat}/K_m for polysulfide generation are essentially equal for CARS and CSE. What is the basis for arguing that that CARS is the more important source of polysulfide generation?

The manuscript contains a lot data showing that persulfide generation by CARS is PLP dependent, and that mutations of the putative lysine attachment points diminish this activity. In view of the fact that the SepCysS enzyme which helps produce cysteinylated-tRNA in the Archaea is also dependent on PLP, this is a compelling result. To help the reader appreciate this scenario, the paper would benefit from the inclusion of a chemical mechanism to explain how the chemistry of this process works. Other questions with regard to this mechanism include;

- (i) When lysines such as K73 and K266 are mutated, is there a commensurate decrease in the extent of modification by PLP? The result that substitution of the lysines in the KMSKS loop have minimal effect on the ability of CARS to support protein synthesis is considerably at odds with the effects of similar substitutions in other class I ARSs.
- (ii) The authors use relatively high concentrations of PLP to observe the modified enzymes in vitro (order of millimolar). Given that intracellular concentrations of PLP are much lower, how confident are the authors of the biological significance of modification by PLP?
- (iii) To what extent are the persulfidation activities of other ARS dependent on PLP modification? Did the authors test whether PARS, HARS, and GARS are similarly stimulated by PLP?

A few additional questions concern the relationship of CARS function to mitochondrial function.

- (i) how is the production of –CysSSH affected by the administration of mitochondrial drugs such as 2,4 dinitrophenol and rotenone? These are more direct inhibitors of electron transport than ethidium bromide, which affects mitochondrial DNA.

Finally, there are some additional minor issues:

Page 4 with the first full paragraph, it is unclear from reading the text along what proteins are

being referred to. While the figure legend clarifies that ADH5 and GAPDH are being studied, this should be noted in the text.

Page 5, they list prolyl-ARS twice in their list.

Page 9, line 4 of discussion: "mitochondrium" should be "mitochondrion".

comment on statistics: there appears to be robust statistical support for key findings.

Reviewer #2 (Remarks to the Author):

Summary

Cysteine persulfidation has recently appeared as a pervasive mode of protein modification. Persulfidation of the Cys residue is believed to occur through oxidation of H₂S by mitochondrial sulfide oxidoreductase or other such enzymes, with H₂S being itself produced from cysteine or homocysteine in a reaction catalyzed by cystathionine beta synthase and cystathionine gamma lyase. In this paper the authors propose the provocative idea that cysteine persulfidation is directly produced from cysteine by cysteinyl-tRNA synthase and others ARSs, and more specifically that aminoacyl tRNA synthetase has PLP-dependent persulfide synthase activity efficiently producing Cys-persulfides. Authors also propose that Cys-persulfide is incorporated into nascent polypeptides during translation provided the use of a tRNA loaded with a Cys-persulfide. Lastly it is shown that in mammalian cells mitochondrial CARS2 function is linked to mitochondrial morphology and bioenergetics, and more specifically that CARS2-dependent generation of persulfides leads to formation of H₂S (the reverse of what is currently believed) through reduction by electrons emanating from the ETC. Hence persulfides produced in mitochondria sustain the activity of the ETC.

Major comments

Strength

If indeed true, the numerous authors' conclusions would constitute a major discovery in the field.

Weakness

This paper is extremely dense, with six busy figures, two tables and 24 supplementary figures, and this is because the paper spread in many directions at the risk of not fully convincing on each of the different points raised, as will be discussed below. As a result of the extended amount of data, experiments are only barely explained, authors use at many places conclusion statements without clear demonstration/explanation, which force the reader to guess the authors

experimental strategy and results, and make the paper highly confusing to read. As a result, this reviewer was not able to fully evaluate this work, because of these facts.

1. As first advice to improve the paper, we suggest the authors carefully chose the experiments that are critically important to establish their point. This will allow them, with a smaller number of figures to provide rationale for each experiment and explain what they are actually showing. It is clear from this reviewer's standpoint that many figures and experiments are not necessary, only confusing the reader.

2. The entire study is based on the authors' finding that cysteine persulfide PSSP or PSS react both as an electrophile (which is the conventional accepted notion) and as a nucleophile. Therefore, PSS(n)P is shown to be degraded both by a thiol reductant (which is conventional), but also reacts with electrophiles to generate different alkylated species of the type PSSS-electrophile. This would be a very new reaction; however, the mechanism of this reaction is never discussed really, and although the results of the experiments shown in suppFig1 is compatible with the authors' conclusion, there are other possible explanations. This reviewer in fact raises the question of whether the nucleophilic reactivity of a PSS is possible? If indeed true, cleavage of the persulfide should generate a compound of the type PSS+, which has never been shown before and if formed should have a high reactivity: what is the fate of this compound is not addressed: it probably becomes a PSS-OH, which is also very reactive towards thiols in particular. It is possible that the solutions used are not pure, and are contaminated with PS or PSS. Alternatively, the PSSS is unstable and degraded in solution prior to react with the electrophile: for instance, the pH effect seen in sup fig 1, of stimulating the reaction might be the consequence of the degradation of the PSSS by attack by a hydroxide anion. Chromatograms/mass spectrograms of the persulfide solutions, of all reactants prior to reaction and of intermediate reaction products must be shown. Polysulfides stability at different pH, in the absence and presence of different concentrations of oxygen should be investigated. Based on the reaction product observed, authors should come out with a plausible model of their reaction mechanisms.

3. The PMSA assay of supplementary fig 4 is surprising. The data would thus show that for instance, in ADH5, there all the enzyme 15 Cys residues become per/polysulfided, including those that coordinate zinc: is that possible? As the gel shows, such a modification would occur homogeneously in all ADH5 molecules, which is also very surprising. Furthermore, even more surprising this modification would be exactly the same using recombinant or endogenous proteins. MS analysis of the entire protein should be performed in order to confirm the data observed on gels.

4. That EcCARS catalyzes the loading of a persulfided Cys on tRNA is surprising but believable. However, the strategy of suppig8 used to demonstrate this fact is not clear: in the scheme, it is

for instance shown that the tRNA is removed, but how this occur? In the text authors mention the presence of a tRNA loaded with a cys persulfide, but this species is only deduced indirectly and never demonstrated by MS to occur. In fact, an alkylating agent should generate the different Cys-Sn species shown in supfig8.

5. That CARS catalyzes a sulfur transfer reaction is also believable. However, sulfur transfer is not a simple reaction: cysteine desulfurase use PLP as cofactor. Here authors show that PLP improves the desulfuration reaction, but the reaction also works prior to adding PLP. Is PLP bound to the recombinant enzyme prior to adding PLP? In sup fig15, this question is not answered but jjust that PLP would bind to specific Lys residues. PLP is required as a prosthetic group to make cysteine out of serine and selenocysteine from serine as well, but a sulfur donor is required for these reactions. Is the authors recation shown in the paper related to these reactions? In addition, the strategy used to make the point in sup fig9 should be similar to the one use in the previous figure: but the differences in the experimental system are not explained. It is extremely hard to get convinced of the result in such a confusing context. When metioning a tRNA, it is usually written tRNACys and not Cys-tRNA as the authors did: this is confusing.

6. The 34S transulfidation experiment of sup fig 12 uis hard to follow and is not carefully explained

4. The experiments in 4 h, e, j would indicate that CSE and CBS have an important contribution in persulfide formation? Which would contradict the overall message of the paper.

Reviewer #3 (Remarks to the Author):

Unlike reactive oxygen and nitrogen species the role of reactive sulfur species in cellular signaling, metabolism and redox homeostasis is underappreciated. The manuscript by Akaike et. al. entitled “Aminoacyl-tRNA synthetases govern protein polysulfidation and mitochondrial bioenergetics” shows that aminoacyl-tRNA synthetases (ARS) have persulfide synthase activity. The authors identified CARS2 (cysteinyl-tRNA syntethase) as a major contributor of protein polysulfidation in the cell. The modified amino acids are incorporated into de novo synthesized protein nascent chains. The authors aimed to uncouple the catalytic function in tRNA aminoacylation from the production of cysteine persulfidation. Since CARS2 is a mitochondrial enzyme the authors went on to describe several mitochondrial deficiencies caused by the lack of this activity. The concept of CARS being responsible for cys persulfide activity and sulfidation being introduced co-translationally to proteins is of high interest. However, some of the findings

or their interpretations in the manuscript are hard to reconcile with the basic cell biology knowledge.

First, the mitochondrial protein synthesis system produces just a few proteins (13 in human), whereas the mitochondrial proteome is estimated to be more than 1000 proteins, but they are all synthesized in the cytosol and subsequently transported. CARS2 as a mitochondrial protein (the references are missing here) should only modify these 13 proteins synthesized in mitochondria and not the other mitochondrial proteins or cellular proteins. Thus, it is very hard to understand how a mitochondrial aminoacyl-tRNA synthetase can lead to persulfidation of the large part of the mitochondrial or cellular proteome and not only these 13 mitochondrial translation products if they are doing so co-translationally. What would be the scenario behind it? What is the location of CARS2? What is the cellular distribution of sulfidated proteins (at least mitochondria vs the rest of the cell)? How does the lack of CARS1 and 2 affect cellular sulfidation in terms of localization?

Furthermore, the way to use various mutants defective in persulfidation vs aminoacylation that are introduced to CARS2-KO is a good direction based on their differential activities determined in vitro for a bacterial system and partially confirmed in the cellular system. However, a major question that remains to be determined experimentally is what are the effects on mitochondrial translation. The mutants specific to sulfidation activity should suppress the translation defect of KO to a wt level.

Along the lines, if CARS2 would be modifying nascent chains by sulfidation during their synthesis being a major contributor to the global cellular persulfidation, it is also critical to check the efficiency of cytosolic translation in CARS2 vs CARS1 deficient cells.

Without thorough investigation of the points above followed by careful interpretations the manuscript (at least its biological part) seem to be much too preliminary.

Other points:

1. To convincingly show that the cysteinyl-tRNA synthetase is a major contributor to protein persulfidation, a direct comparison of the amount of endogenous modified proteins dependent on cystathionase and cystathionine β -synthase or the cysteinyl-tRNA synthetase would be helpful (even using down-regulation approaches).
2. Recent data of Coughlin et. al., JMG, 2015 already correlated mutations in CARS2 with deficiencies in respiratory chain complexes and associated mitochondrial disease in a patient study. Akaike and colleagues should acknowledge this study in their manuscript.
3. Please, avoid bold statements such as the one in the abstract about evolution of the central dogma in molecular biology.

Reply to the Reviewers' comments:

To Reviewer #1:

Reviewers' comments:

Reviewer #1 (Remarks to the Author):

General Comments:

This paper addresses the biological occurrence, biosynthesis, and potential function of cysteine hydropersulfides in biological systems. Previous reports by these authors have suggested the importance of compounds containing reactive persulfides (-SSH) as potential mediators of antioxidant effects. Currently, details are lacking regarding the enzymes principally responsible for their synthesis, and how these effects are mediated. One school of thought holds that the enzymes cystathionase (CSE) and cystathionine beta synthase (CBS) readily convert cysteine to cysteine hydropersulfide. However, these authors argue that the kinetic parameters of these reactions may not be sufficient to normal cellular physiology, and that -SSH is likely being produced by other sources. Other biosynthetic routes of -CSSH formation remain to be identified.

This manuscript contains a huge amount of information, and not all findings can be summarized here. Briefly, the central finding of this manuscript is that cysteinyl tRNA synthetase may be a major source of persulfidated compounds in cells, and this activity may be particularly important to support mitochondrial function.

(partially omitted)

Overall, this manuscript opens up a novel and potentially highly significant area of biochemistry whereby a standard enzyme of translation produces a new translational modification of proteins, with many implications for regulation of mitochondrial function. The wealth of information provided in this paper provides support for many of the principal conclusions of the paper. The provocative nature of this study does raise some questions and concerns that should be addressed.

Response:

We are very grateful and honored for such an insightful comment on our present work on CARS and reactive persulfides such as CysSSH.

To verify the idea that CARS2 truly functions as a major source of endogenous reactive persulfides, especially CysSSH, we sought to develop *Cars2* knockout (KO) mice and just recently succeeded in doing so in our laboratory. Because we observed a very clear phenotype in terms of persulfide production in mice, we decided to incorporate and highlight the *in vivo* results that we obtained with these *Cars2* KO mice (please see new Figs. 5 - 7, and new Supplementary Fig. 19 - 21 and their legends: P. 2, L. 8 - 11; P. 8, L. 14 - P. 9, L. 28; P. 12, L. 5 - 25; P. 16, L. 37 - P. 17, L. 20).

Nevertheless, we carefully considered the possibility that the paper included too much information, which may make understanding the paper difficult for readers. Accordingly, we did carefully editing so as to streamline the paper, which we believe should address the general concerns that you raised here.

Questions and Concerns:

Specific comments #1

A somewhat broad question concerns the relative importance of the two different pathways for generation of persulfidated cysteines in the client proteins. Two potential mechanisms are presented: (A) incorporation into proteins via aminoacylation and standard protein synthesis, as suggested by the data from Figure 1c,d,e and (B) modification of target proteins via a PLP dependent mechanism, as suggested by the

data in Figure 2 and supp Fig 4. Depending on which of the two mechanisms predominates, there are a number of important predictions and issues.

If mechanism A predominates, this raises the question of a balance between normal cysteine and –SSH incorporation is achieved, and how wholesale incorporation of –SSH in all positions and in all proteins is avoided/regulated. This point is not really highlighted.

Response:

We know from our PUNCH-PsP approach (new Fig. 1c and Supplementary Fig. 10; main text, P. 4, L. 37 – P. 5, L. 7; new Supplementary Information, P. 43, L. 38 – P. 44, L. 29) that CysSSH is evenly and equally incorporated into nascent peptides, which are expected to be cleaved from or edited to the appropriate size of polysulfide chains by the thioredoxin (Trx)/Trx reductase system, as discussed in the original text and reported by Peter Nagy's group (Ref. 4). We now highlight this possible mechanism in the revision better than in the earlier version of our manuscript: new Figs. 7c, 8e, and 10; P. 9, L. 21 – 28; P. 10, L. 9 – 13; P. 13, L. 33 – 35.

Although some *E. coli* ARSs other than EcCARS showed appreciable CysSSH-producing activity, because the catalytic mechanism of other ARSs remains unclear and definitely needs more extensive study for clarification, we decided to remove these ARS data (old Supplementary Fig. 14 and its legend) from our revision.

Moreover, because LC-MS/MS analysis with *Cars2* KO mice indicated that mammalian CARS2 mainly contributes to persulfide production in whole cells and tissues, we believed that our current study should focus more on the biological relevance and physiological functions of mammalian CARS2 as a newly discovered cysteine persulfide synthase (CPERS) *in vivo*. Please see the data and related information on our *in vivo* study using *Cars2* KO mice on P. 2, L. 8 – 11; P. 8, L. 14 – P. 9, L. 28; P. 12, L. 5 – 25; P. 16, L. 37 – P. 17, L. 20 in the new text and in new Figs. 5-7; Supplementary Fig. 19-21.

Specific comments#2:

Based on the known amino acid specificity of ARSs, it is difficult to imagine other ARSs being able to carry out this function. This point is not really highlighted.

Response:

As just mentioned, although we reproducibly obtained persulfide synthase activity with various ARSs from *E. coli*, because its catalytic mechanism is still obscure, we omitted ARSs data (old Supplementary Fig. 14 and its legend in the old manuscript).

As a notable finding, we verified by our current *in vivo* study with *Cars2* KO mice that CARS2 primarily produces CysSSH and all other sulfide derivatives, including HS⁻ (please see new Figs. 5 – 7 and Supplementary Fig. 19 – 21 and their legends). At least in the mammalian system, CARS2 was a major source of persulfides; for the *E. coli* enzyme, however, this may not be the case. To streamline the content of our paper, which this Reviewer mentioned, we therefore decided to focus on the biological functions of CARS2 because of its predominance in endogenous persulfide formation in cells and *in vivo* in mammals.

Specific comment #3:

Other minor question regarding this mechanism are: (i) did the authors demonstrate that this aminoacylation reaction is ATP dependent; and ...

Response:

Yes, the aminoacylation itself depends on ATP, but the persulfide production did not, as described on P. 5, L. 10 – 12.

Specific comment #4:

(ii) how does their experimental aminoacylation procedure (Supplementary page 12 and 41) rigorously limit the detection of derivatized free amino acid?

Response:

After the aminoacylation reaction, Cys-tRNA^{Cys} and CysSSH/SSSH-bound Cys-tRNA (Cys-tRNA^{CysSSH/SSSH}) were isolated by trapping with a cotton wool filter, and subsequently the free Cys and CysSSH/SSSH were completely removed by repeatedly washing the tRNA-containing cotton wool with 10% trichloroacetic acid + 70% ethanol. This explanation originally appeared on P. 41 in old Supplementary Information, but we provide a more careful explanation in the new Supplementary Fig. 8 and its legend and in Supplementary Information: P. 40, L. 32 – P. 41, L. 6.

Specific comment #5:

If mechanism B predominates, there are other questions that are unresolved. First, there is the claim CysRS is much more efficient producer of –CysSSH than either CSE or CB. However data in Supp Fig13 and Supp Table 2 indicate that the k_{cat}/K_m for polysulfide generation are essentially equal for CARS and CSE. What is the basis for arguing that that CARS is the more important source of polysulfide generation?

Response:

We would like to ask this Reviewer to consider the following points. Although the k_{cat}/K_m values are almost equal for these enzymes, CSE/CBS and CARS utilize different substrates, i.e., only cystine (but not cysteine) and cysteine (but not cystine) for CysSSH production, respectively. In addition, the physiological concentrations of cystine are in the low- or sub-micromolar range, which is far lower than its K_m , 200 μ M for CSE. As stated herein and recently described (new Supplementary Table 3 and new Supplementary Fig. 12 and its legend; P. 5, L. 22 – P.6, L. 3 in the new text; Refs. 3, 20), CSE cannot utilize cysteine for its production of persulfide. This fact thus indicates that the cystine/CSE reaction could not compete successfully with the reactions with other enzymes metabolizing cystine and substances such as GSH, which interact chemically with cystine. The intracellular cysteine concentration is reportedly about 100-1000 μ M in cells and major organs (Ref. 3 and our current study), which is much higher than the K_m of CARS. This biochemical insight strongly supports the idea that CARS is the more important source of polysulfide generation compared with other enzymes such as CSE and CBS.

To respond to similar comments by Reviewers 2 and 3, we provided solid pieces of evidence in this revision, which elucidates the predominant contribution of CARS2 to endogenous persulfide production. Results from a gene silencing study of CSE/CBS expressed in cultured cells clearly showed that CysSSH production that was apparently regulated by CSE/CBS depended solely on CARS2 expression in HEK293T cells. In other words, CARS2 likely controls CysSSH production in the whole cell. This idea is clearly verified by our *in vivo* study with *Cars2* KO mice that we just developed, as mentioned above.

We added these comments to the revision: Supplementary Fig. 18, and P. 8, L. 5 – 13.

Specific comment #6:

The manuscript contains a lot of data showing that persulfide generation by CARS is PLP dependent, and that mutations of the putative lysine attachment points diminish this activity. In view of the fact that the SepCysS enzyme which helps produce cysteinylated-tRNA in the Archaea is also dependent on PLP, this is a compelling result. To help the reader appreciate this scenario, the paper would benefit from the inclusion of a chemical mechanism to explain how the chemistry of this process works.

Response:

Thank you for this comment. PLP functions as an important coenzyme in diverse enzymatic reactions including amino acid racemizations, eliminations, additions, and substitutions. In virtually all PLP-dependent reactions, the aldehyde group of PLP forms an imine (Schiff base) linkage with a lysine side chain on the enzyme. In the cysteine desulfurase reaction, the amine nitrogen of the substrate cysteine replaces the enzyme lysine nitrogen in the imine linkage (imine exchange) to form a cysteine-PLP adduct (Ref. 43). Abstraction of a proton facilitates the formation of a cysteine-quinoid intermediate, and sulfur is subsequently released from the intermediate to the acceptor cysteine residue to form persulfide (Ref. 43). PLP-dependent sulfur transfer has also been identified in *Archaea* aminoacyl-tRNA synthetase. In methanogenic *Archaea*, for example, Sep-tRNA:Cys-tRNA synthetase (SepCys-tRNA synthetase) catalyzes sulfhydration of tRNA-bound *O*-phosphoserine (Sep-tRNA) to form cysteinyl-tRNA^{Cys} in a PLP-dependent manner (Ref. 44). Sep-tRNA binds to SepCys-tRNA synthetase and forms an imine linkage, after β -elimination of a phosphate group to form dehydroalanyl-tRNA^{Cys}, whose double bond is attacked by persulfide that transferred from another cysteine desulfurase such as IscS, which is also well known as a PLP-dependent enzyme, to finally form a Cys moiety on tRNA (i.e., Cys-tRNA^{Cys}) (Ref. 44).

For CARS-mediated persulfide formation, therefore, we propose that one sulfur moiety is eliminated from a cysteine thiol and is transferred to another cysteine (CysSH) thiol moiety to form cysteine persulfide (CysSSH) in a PLP-dependent fashion, as catalyzed by a single CARS enzyme to produce CysSSH. Indeed, we verified this enzymatic reaction mechanism via a stable isotope ³⁴S tracer analysis of the sulfur transfer reaction catalyzed by CARS, as new Supplementary Fig. 11 shows.

The revised text now includes this brief discussion : P. 13, L. 37 – P. 14, L. 16; Supplementary Information, P. 41, L. 24 – 29.

Specific comment #7:

Other questions with regard to this mechanism include;

(i) When lysines such as K73 and K266 are mutated, is there a commensurate decrease in the extent of modification by PLP? The result that substitution of the lysines in the KMSKS loop have minimal effect on the ability of CARS to support protein synthesis is considerably at odds with the effects of similar substitutions in other class I ARSs.

Response:

As this Reviewer pointed out, there may be a “commensurate” change in the binding capacity and/or stability of PLP as a result of mutation of Lys (K) residues among the four Lys residues, because each single Lys mutation at the KIIK/MKSK motifs affects all CysSSH synthesis activity. One possible explanation is that PLP may need multiple Lys residues, rather than a single Lys binding, as expected by the computational structural analysis (Fig. 3a), for stable binding and catalytic activity. PLP-dependent catalytic activity may also be stabilized by such a multiple Lys binding, because the highly nucleophilic nature of CysSSH and CysS-(S)_n-H effectively

produced by the enzyme CARS should readily interfere with the electrophilic aldehyde group of PLP to form an imine (Schiff base) linkage on the Lys residues, which would cause instability of the catalytic activity of PLP bound with these particular K residues of CARS.

With regard to the KMKS motif that is conserved among class I ARS, one of the class I enzymes, arginyl-tRNA synthetase (ArgRS), lacking a canonical KMSK sequence, still has a conserved Lys residue in the upstream of the HIGH motif, which was suggested to be an alternative for the canonical class I lysine in the KMSK sequence (Ref. 45). This result indicates that lysine-containing sequences other than the KMSK sequence may exist and may function as important residues that may be involved in catalysis of the class I ARSs.

It is interesting that *E. coli* CysRS (EcCARS) has two KMSK motifs (¹⁸⁵KMSK¹⁸⁸ and ²⁶⁶KMSK²⁶⁹), and the two lysines of the first ¹⁸⁵KMSK¹⁸⁸ motif located at the N-terminal side of EcCARS are conserved well in human and mouse CARS2 (²³³KAAK²³⁶ and ²¹⁸KAAK²²¹, respectively). In fact, our computational analysis with the PyMOL molecular graphics system showed that this additional ¹⁸⁵KMSK¹⁸⁸ motif in EcCARS is located in the vicinal area of the active site zinc ion coordinated by Cys residues (within 20 Å, closer than ²⁶⁶KMSK²⁶⁹) and the KIIK and KMSK motifs (also within 20 Å) as well (new Fig. 3a).

All these data therefore suggested that some particular Lys residues other than the KMSM motif may constitute a part of the ATP-binding site at the HIGH motif and may thus contribute to ARS catalytic activity for aminoacyl-tRNA synthesis, as supported by the EcCARS three-dimensional structure.

We briefly discuss this interpretation in our revision: P. 14, L. 17 – P. 15, L. 4. Please note that our three-dimensional structure modeling supported this interpretation of the commensurate effect of these KXXX motifs, which are indeed located in a vicinal distance (each within 10-20 Å), as illustrated and highlighted in the three-dimensional structure in new Fig. 3a. We also comment on this issue on P. 6, L. 22 – P. 7, L. 8 in the new text.

Specific comment #8:

(ii) The authors use relatively high concentrations of PLP to observe the modified enzymes in vitro (order of millimolar). Given that intracellular concentrations of PLP are much lower, how confident are the authors of the biological significance of modification by PLP?

Response:

We usually used PLP at 50 μM or less when we added it to biochemical enzymatic reactions. To our best knowledge, 10-100 μM concentrations are widely used for biochemical reactions in general for PLP-dependent enzymes, e.g., CSE/CBS. We used PLP concentrations higher than 0.1 mM for Q-TOF-MS analysis to identify PLP-binding sites in recombinant EcCARS (new Supplementary Fig. 13). According to the method previously reported (new Supplementary Ref. 17, added to the text of the new Supplementary Information; P. 44, L. 31 – P. 45, L. 2), to effectively achieve PLP binding, we need relatively high concentrations of PLP (in fact, Supplementary Ref. 17 used saturated PLP for protein treatment), compared with the regular biochemical assay just mentioned, because the reductive treatment (+ urea denaturation), which is necessary for protein alkylation before tryptic digestion, seems to affect the PLP-binding capacity for protein Lys residues. Therefore, such a high millimolar concentration of PLP was applied just for this analytical purpose, without physiological relevance. Nevertheless, it is important to point out that biologically functioning Lys

residues that we discovered as catalytic sites for the persulfide production include K73/266/289, which are bound to endogenous PLP (i.e., without addition of PLP) and which bind with a PLP concentration as low as 100 μ M, which was used for treatment of EcCARS applied to Q-TOF analysis.

We provide this explanation with the new reference citation in the Supplementary Information P. 44, L. 31 – P. 45, L. 2.

Specific comment #9:

(iii) To what extent are the persulfidation activities of other ARS dependent on PLP modification? Did the authors test whether PARS, HARS, and GARS are similarly stimulated by PLP?

Response:

We must address this important question in the future. As I mentioned above, to streamline the text, we decided to remove the information related to ARSs. I regret any confusion related to this matter.

Specific comment #10:

A few additional questions concern the relationship of CARS function to mitochondrial function.

(i) how is the production of –CysSSH affected by the administration of mitochondrial drugs such as 2,4 dinitrophenol and rotenone? These are more direct inhibitors of electron transport than ethidium bromide, which affects mitochondrial DNA.

Response:

This issue is also critical and we would like to clarify it in our future studies, because we must think carefully about the various ETC complexes, whose inhibition may indirectly affect functions of the whole cell. Nevertheless, in response to this comment, we performed an additional experiment to investigate the effects of mitochondria ETC inhibition by antimycin A, a complex III inhibitor; we used LC-MS/MS analysis to determine CysSSH and its reductive sulfide derivatives, such as HS⁻ and thiosulfate. As new Fig. 9c-e shows, CysSSH can indeed serve as an electron acceptor for the ETC, possibly complex IV, similar to molecular oxygen (O₂). We added this critical and rather new finding, along with a modified scheme for persulfide-mediated mitochondrial functions that we proposed (new Figs. 9c-i and 10b), and related information to the revised manuscript: P. 10, L. 32– P. 12, L. 2; Supplementary Information, P. 47, L. 20 – P. 48, L. 4.

Other minor comment #1:

Finally, there are some additional minor issues:

Page 4 with the first full paragraph, it is unclear from reading the text along what proteins are being referred to. While the figure legend clarifies that ADH5 and GAPDH are being studied, this should be noted in the text.

Response:

My apologies for this awkward explanation of the results in old Fig. 1a and old Supplementary Fig. 6. The new text now incorporates the exact proteins analyzed (ADH5 and GAPDH): P. 4, L. 18 – 23 (also please see new Supplementary Information, P. 42, L. 32 – 41.

Page 5, they list prolyl-ARS twice in their list.

Response:

Thank you for pointing out this error, which we corrected accordingly.

Page 9, line 4 of discussion: "mitochondrium" should be "mitochondrion".

Response:

Thank you. We corrected the spelling and now use "mitochondrion": P. 12, L. 26.

Comment on statistics: there appears to be robust statistical support for key findings.

Response: Thank you for your thoughtful evaluation of our statistical analysis, which we in fact reinforced in the revision.

In addition to our alterations just mentioned and described in this response to the Reviewers, we made a few passages of modifications to improve the content of our work. These changes include the following:

1. We originally included 6 main figures but now have 10; we added 3 new figures, including, for example, data on *Cars2* KO mice, and we somewhat changed the order of their presentation.
2. We transferred certain main figures to the Supplementary Information (new Supplementary Figs. 7a, 24b, 25b), as well as deleted ARS and MeHg data (old Figs. 4h-j, Supplementary Figs. 14, 24, and their related descriptions).
3. We modified and improved the scientific and illustrative quality of the schematics (e.g., in new Figs. 3a, 4, 8, and 10), according to revisions that we made to respond to the Reviewers' comments.
4. The scientific validity and reproducibility of some original experiments were qualified rigorously by repeating these studies: please see new Figs. 2a,e and 4c,g; new Supplementary Fig. 12a; new Supplementary Table 3.
5. The experiments with *CARS2* KO cells and their results and interpretation were explained and interpreted in the revised text much more carefully and in detail than in the old text (new Figs. 4 and 8; new Supplementary Fig. 15; P. 7, L. 13 – P. 8, L. 4).

We also made minor modifications in the main text to clarify our message. Please use the marked-up versions to check our revision.

In summary, we would like to thank this Reviewer so much for extremely careful and insightful comments, which were of great help for improving our work.

Reply to the Reviewers' comments:

To Reviewer #2:

Reviewer #2 (Remarks to the Author):

Summary

Cysteine persulfidation has recently appeared as a pervasive mode of protein modification. Persulfidation of the Cys residue is believed to occur through oxidation of H₂S by mitochondrial sulfide oxidoreductase or other such enzymes, with H₂S being itself produced from cysteine or homocysteine in a reaction catalyzed by cystathionine beta synthase and cystathionine gamma lyase. In this paper the authors propose the provocative idea that cysteine persulfidation is directly produced from cysteine by cysteinyl-tRNA synthase and others ARSs, and more specifically that aminoacyl tRNA synthetase has PLP-dependent persulfide synthase activity efficiently producing Cys-persulfides. Authors also propose that Cys-persulfide is incorporated into nascent polypeptides during translation provided the use of a tRNA loaded with a Cys-persulfide. Lastly it is shown that in mammalian cells mitochondrial CARS2 function is linked to mitochondrial morphology and bioenergetics, and more specifically that CARS2-dependent generation of persulfides leads to formation of H₂S (the reverse of what is currently believed) through reduction by electrons emanating from the ETC. Hence persulfides produced in mitochondria sustain the activity of the ETC.

Response:

First, we are grateful and honored that this Reviewer understood our present findings on the unique and novel mechanism of reactive persulfide formation and its critical function in mitochondria and protein polysulfidation.

Comment #1:

[Major comments]

Strength

If indeed true, the numerous authors' conclusions would constitute a major discovery in the field.

Response-1:

We truly appreciate this encouraging comment, which emphasizes the novelty of our present findings on CARS and its functions mediated by per/polysulfides in mitochondria.

Weakness

This paper is extremely dense, with six busy figures, two tables and 24 supplementary figures, and this is because the paper spread in many directions at the risk of not fully convincing on each of the different points raised, as will be discussed below. As a result of the extended amount of data, experiments are only barely explained, authors use at many places conclusion statements without clear demonstration/explanation, which force the reader to guess the authors experimental strategy and results, and make the paper highly confusing to read. As a result, this reviewer was not able to fully evaluate this work, because of these facts.

1. As first advice to improve the paper, we suggest the authors carefully chose the experiments that are critically important to establish their point. This will allow them, with a smaller number of figures to provide rationale for each experiment and explain what they are actually showing. It is clear from this reviewer standpoint that many figures and experiments are not necessary, only confusing the reader.

Response-2:

We carefully considered this criticism and agree that our original paper was dense, with too much information, which may somehow confuse readers who may be unable to thoroughly digest the text and data. To address this concern, we carefully edited the content to streamline the paper.

As one editorial change, for example, we decided to remove ARS data and related information (old Supplementary Fig. 14 old text, P. 5, L. 22 – 28). Although we reproducibly obtained persulfide synthase activity with various ARSs from *E. coli*, we believe that the catalytic mechanism of ARSs other than CARS remains unclear and needs more extensive study,

More important, we successfully developed *Cars2* knockout (KO) mice (as shown in new Figs. 5-7, and new Supplementary Figs. 19-21 and their legends: P. 2, L. 8 – 11; P. 8, L. 14 – P. 9, L. 28; P. 16, L. 37 – P. 17, L. 20). LC-MS/MS analysis identified mammalian CARS2 as truly contributing to most cell and tissue persulfide production, a finding that strongly supported the biological relevance and physiological functions of mammalian CARS2 as a major source of cysteine persulfide synthesis *in vivo*, as we indeed discovered.

Comment #2:

2. The entire study is based on the authors finding that cysteine persulfide PSSP or PSS react both as an electrophile (which is the conventional accepted notion) and as a nucleophile. Therefore, PSS(n)P is shown to be degraded both by a thiol reductant (which is conventional), but also reacts with electrophiles to generate different alkylated species of the type PSSS-electrophile. This would be a very new reaction; however, the mechanism of this reaction is never discussed really, and although the results of the experiments shown in suppFig1 is compatible with the authors' conclusion, there are other possible explanations. This reviewer in fact raises the question of whether the nucleophilic reactivity of a PSS is possible?

Response:

Our apologies for an incomplete narrative and lack of an appropriate statement in the text and Supplementary section. Since our paper focused more on the biological aspect of CARS rather than the chemistry of the polysulfide reaction with electrophiles, we actually omitted a large part of the information on polysulfide chemistry in the original manuscript, most of which is actually known.

In fact, the dual electrophilic-nucleophilic character of hydropersulfides is well documented (the anionic RSS^- species being nucleophilic and the protonated $RSSH$ species possessing electrophilic properties akin to disulfides, $RSSR$). This Reviewer is correct in stating that the nucleophilic character of polysulfides such as RSS_nR ($n > 1$) is not well known. However, this reaction is not a new one. Several previous reports demonstrated the ability of a trisulfide species to react with numerous electrophiles. For example, Fletcher and Robson reported that thiocystine (cysteine trisulfide, Cys-SSS-Cys) readily reacted with electrophilic halogens (e.g., Br_2), which resulted in cleavage of the S-S bond (Ref. 24). A review by Parker and Kharasch also discussed numerous examples of the electrophilic cleavage of the S-S bond in disulfides by electrophilic reagents such as protons, sulfenium ions, and halogens (Ref. 25). More recently (and directly relevant to our studies), Abdolrasulnia and Wood reported that Cys-SSS-Cys reacted readily with iodoacetic acid (a well-established thiol-modifying agent) to ultimately give carboxymethylthiocystine (Cys-SS- CH_2COOH) (Ref. 26), which is consistent with the idea that a nucleophilic sulfur atom of the trisulfide reacted with the electrophilic iodoacetic acid species and led to S-S bond cleavage. Thus,

ample precedence for the nucleophilic character of polysulfides exists, and it is not surprising that electrophilic modification of these species is possible.

We provide this information in the new text: P. 3, L. 35 – 37; P. 13, L. 6 – 23.

Nevertheless, we believe that the old Supplementary Fig. 2 may not have clearly illustrated such a unique dual electrophilic-nucleophilic character of CysS-(S)_n-H, and we decided to provide a new Supplementary Fig. 2, which now shows more convincingly that, depending on the electrophilicity, HPE-IAM and NEM demonstrated quite different profiles of formation of individual adducts with CysSH and CysSSH/SSSH. That is, NEM (a strong electrophile) produced mostly CysS-NEM (a simple cysteine adduct), even if the reaction mixture contained mainly cysteine polysulfides (CysSSH and SSSH), whereas HPE-IAM (a weak electrophile) formed adducts with CysSSH/SSSH. Please see the new Supplementary Fig. 2 and its legend.

Comment #3:

If indeed true, cleavage of the persulfide should generate a compound of the type PSS⁺, which has never been shown before and if formed should have a high reactivity: what is the fate of this compound is not addressed: it probably becomes a PSS-OH, which is also very reactive towards thiols in particular. It is possible that the solutions used are not pure, and are contaminated with PS or PSS. Alternatively, the PSSS is unstable and degraded in solution prior to react with the electrophile: for instance, the pH effect seen in sup fig 1, of stimulating the reaction might be the consequence of the degradation of the PSSS by attack by a hydroxide anion. Chromatograms/mass spectrograms of the persulfide solutions, of all reactants prior to reaction and of intermediate reaction products must be shown. Polysulfides stability at different pH, in the absence and presence of different concentrations of oxygen should be investigated. Based on the reaction product observed, authors should come out with a plausible model of their reaction mechanisms.

Response:

To support the possible formation of R-SS⁺ (R-SS-OH) species during electrophilic decomposition of GS-(S)_n-SG, we indeed identified a dimeric adduct of sulfenic acids (-SOH) and/or its persulfide form, i.e., -SSOH (data not shown). This finding is quite new and still preliminary, and we will need additional study to fully understand the chemistry of such a unique reaction of dimeric with -SSOH molecular species.

We thoroughly checked the purity of the various polysulfides used, and no reduced thiols, even hydro-per/polysulfides, were contaminated. It seems that GSSSSG, for example, is relatively unstable under alkaline pH and undergoes spontaneous decomposition, possibly with simultaneous production of sulfenic acid, as just mentioned, GSSSSG is not affected by molecular oxygen (not shown here).

However, we hope to report this extensive analysis of persulfide sulfenic acid or the exact chemistry of polysulfide decomposition in the future. We must avoid dense papers, and because this Reviewer kindly recommended that we carefully choose the experiments that are critically important to establish our point, we decided not to include these preliminary data in the revision.

Comment #4:

3. The PMSA assay of supplementary fig 4 is surprising. The data would thus show that for instance, in ADH5, there all the enzyme 15 Cys residues become per/polysulfided, including those that coordinate zinc: is that possible? As the gel shows, such a modification would occur homogeneously in all ADH5 molecules, which is also very

surprising. Furthermore, even more surprising this modification would be exactly the same using recombinant or endogenous proteins. MS analysis of the entire protein should be performed in order to confirm the data observed on gels.

Response:

We are sorry for perhaps confusing this Reviewer. We do not claim that all Cys residues of every mature protein are polysulfidated; our current proposal is that they are polysulfidated initially at a translational level in ribosomes and are then de-polysulfidated by a particular mechanism such as Trx, as recently reported by Nagy's group in *Sci. Adv.* (Ref. 4).

In fact, we already examined several different recombinant and endogenous proteins for polysulfidation, and we indeed found that all proteins are not fully polysulfidated. In other words, the level of polysulfidation varied among the proteins examined. For example, the redox sensor protein Keap1 is less polysulfidated (not shown herein; we hope to report this elsewhere), whereas ADH5 is highly polysulfidated. ETHE1, GAPDH, and Drp1 are also markedly polysulfidated, but less so than ADH5, as we describe here and also reported in our earlier paper (Ref. 13). Even the proteins such as ADH5 and ETHE1 showed reduced polysulfidation with mutation of particular Cys residues (Ref. 13), which seem to be responsible for sustaining the structure and activity of each protein, we suggest that some chemical mechanism, yet unidentified, for example, metal coordinates such as Zn and Fe ions, may exist to maintain the polysulfide moieties in the protein. This idea is supported by a recent paper on an Fe-S protein, published by another group (Ref. 14), which clearly showed that thiols of Cys residues coordinated to iron were polysulfidated. Therefore, for Zinc ligand thiols to be polysulfidated is feasible as long as the sulfide moieties are first incorporated into respective Cys residues during the early translational process and are appropriately sustained.

Because the protein polysulfidation status is maintained by a delicate balance regulated by polysulfidation and depolysulfidation processes, it may not be practical to investigate all proteins expressed in cells for polysulfidation, which should vary depending on the spatial-temporal regulation of polysulfidation in different cells and tissues *in vivo*, and even in their organelles, as we proposed herein.

Nevertheless, we did study the steady state of polysulfidation in cells and *in vivo* by means of LC-MS/MS analysis, which showed a remarkable level, almost 10% Cys polysulfidation, in the whole proteins recovered from cells and tissues *in vivo* (see new Fig. 7 and the legend). It is striking that such a high level of polysulfidation was partly controlled by CARS2: more than 20% of whole cell and tissue protein polysulfidation depended on CARS2 expression, as evidenced by *CARS2* KO HEK293T cells and *Cars2* KO mice *in vivo* and as shown in new Fig. 7. This again confirmed the idea that polysulfidation occurs during both translational and post-translational modification (PTM) processes. In fact, this PTM is mediated by CySSH released extramitochondrially, as described below. Please also see the response to Reviewer #3 Comment #1. We provide a brief discussion of this interpretation in the revision: P. 3, L. 9 – 14; P. 4, L. 37 – P. 5, L. 7; P. 10, L. 1 – 13; P. 13, L. 33 – 35; new Figs. 7, 8e, and 10.

Comment #5:

*4. That EcCARS catalyzes the loading of a persulfidated Cys on tRNA is surprising but believable. However, the strategy of *supgig8* used to demonstrate this fact is not clear: in the scheme, it is for instance shown that the tRNA is removed, but how this occur? In the text authors mention the presence of a tRNA loaded with a cys persulfide, but this*

species is only deduced indirectly and never demonstrated by MS to occur. In fact, an alkylating agent should generate the different Cys-Sn species shown in supfig8.

Response:

Reviewer #1 raised the same question, which we responded to earlier, and we provide our answer here again. Cys-tRNA bound with both Cys and CysSSH (or Cys-(S)_n-H), i.e., Cys-tRNA^{Cys} and Cys-tRNA^{CysSSH(Cys-(S)_n-H)}, were synthesized by EcCARS with Cys and CysSSH (or Cys-(S)_n-H) as substrates and were alkylated with HPE-IAM to form stable Cys and Cys-(S)_n-H adducts of Cys-tRNA^{Cys} and Cys-tRNA^{CysSSH(Cys-(S)_n-H)}. These adducts were then acid-precipitated with 10% trichloroacetic acid, followed by trapping with a cotton wool filter (100 µl), which was subsequently washed twice with 10% trichloroacetic acid (200 µl) and then twice with 70% ethanol (200 µl) to completely remove free Cys and CysSSH/SSSH. Cys-HPE-IAM and Cys-(S)_n-HPE-IAM were dissociated by using alkaline heat hydrolysis of the ester bond of aminoacyl moieties of Cys-tRNA^{Cys} and Cys-tRNA^{CysSSH(Cys-(S)_n-H)}. The hydrolysis was performed in 20 mM Tris-HCl (pH 8.0), which contained known amounts of stable isotope-labeled internal standards, at 70 °C for 15 min, followed by LC-MS/MS analysis, whose spectra now appear in new Supplementary Fig. 8.

This explanation originally appeared on P. 41, but we provide a more careful explanation in the new Supplementary section: P. 40, L. 32 – P. 41, L. 6

Comment #6:

5. That CARS catalyzes a sulfur transfer reaction is also believable. However, sulfur transfer is not a simple reaction: cysteine desulfurase use PLP as cofactor. Here authors show that PLP improves the desulfuration reaction, but the reaction also works prior to adding PLP. Is PLP bound to the recombinant enzyme prior to adding PLP? In sup fig15, this question is not answered but jyust that PLP would bind to specific Lys residues. PLP is required as a prosthetic group to make cysteine out of serine and selenocysteine from serine as well, but a sulfur donor is required for these reactions. Is the authors rection shown in the paper related to these reactions?

Response:

As this Reviewer pointed out, the PLP-dependent sulfur transfer reaction is in general complex. In fact, PLP functions as an important coenzyme in diverse enzymatic reactions including amino acids racemizations, eliminations, additions, and substitutions. In virtually all PLP-dependent reactions, the aldehyde group of PLP forms an imine (Schiff base) linkage with a lysine side chain on the enzyme. In the cysteine desulfurase reaction, the amine nitrogen of the substrate cysteine replaces the enzyme lysine nitrogen in the imine linkage (imine exchange) to form a cysteine-PLP adduct (Ref. 43). Abstraction of a proton facilitates the formation of a cysteine-quinoid intermediate, and sulfur is subsequently released from the intermediate to the acceptor cysteine residue to form persulfide (Ref. 43). PLP-dependent sulfur transfer has also been identified in *Archaea* aminoacyl-tRNA synthetase. In methanogenic *Archaea*, for example, Sep-tRNA:Cys-tRNA synthetase (SepCys-tRNA synthetase) catalyzes sulfhydration of tRNA-bound *O*-phosphoserine (Sep-tRNA) to form cysteinyl-tRNA^{Cys} in a PLP-dependent manner (Ref. 44).

Sep-tRNA binds to SepCys-tRNA synthetase and forms an imine linkage, after β-elimination of a phosphate group to form dehydroalanyl-tRNA^{Cys}, whose double bond is attacked by persulfide that transferred from another cysteine desulfurase such as IscS, which is also well known as a PLP-dependent enzyme, to finally form a Cys moiety on tRNA (i.e., Cys-tRNA^{Cys}) (Ref. 44). For CARS-mediated persulfide formation, therefore, we propose that one sulfur moiety is eliminated from a cysteine (CysSH) thiol and is transferred to another cysteine thiol moiety to form cysteine persulfide (CysSSH)

in a PLP-dependent fashion, as catalyzed by a single CARS enzyme to produce CysSSH. Indeed, we verified this enzymatic reaction mechanism via a stable isotope ^{34}S tracer analysis of the sulfur transfer reaction catalyzed by CARS, as new Supplementary Fig. 11 shows.

The revised text now includes this brief discussion: P. 13, L. 37 – P. 14, L. 16.

Comment #7:

In addition, the strategy used to make the point in sup fig9 should be similar to the one use in the previous figure: but the differences in the experimental system are not explained. It is extremely hard to get convinced of the result in such a confusing context. When mentioning a tRNA, it is usually written tRNACys and not Cys-tRNA as the authors did: this is confusing.

Response:

I believe that the protocols for Supplementary Figs. 8 and 9 are described in the old Supplementary Methods (P. 41-42). However, because of our paper's dense and incomplete information, the Reviewer may have missed our explanation. In any event, the purpose of the study illustrated in Supplementary Fig. 9 is the same as that shown in a previous figure (Supplementary Fig. 8), with the protocols explained in detail in the Response to Comment #4 above. However, the method used here differs from that for Supplementary Fig. 9. Supplementary Fig. 8 shows HPE-IAM adducts of CysSSH and Cys-(S)_n-H released from Cys-tRNA^{Cys} and Cys-tRNA^{CysSSH(Cys-(S)_n-H)} by heat-alkaline treatment, which cleaves the ester bond formed between the –COOH of Cys and the –OH of ribose (adenosine) of Cys-tRNA^{Cys} and Cys-tRNA^{CysSSH(Cys-(S)_n-H)}. For Supplementary Fig. 9, however, Cys-tRNA^{Cys} and Cys-tRNA^{CysSSH(Cys-(S)_n-H)} formed by CARS were first stabilized by acetylation followed by digestion with RNase to obtain a free adenosine bound with HPE-AM CysSSH and Cys-(S)_n-H adducts, as Supplementary Figure 9a illustrates. Therefore, although the molecules were the same, the detection method used for Supplementary Fig. 9 was much more specific for the Cys and Cys-(S)_n-H residues formed on Cys-tRNA than that used for Supplementary Fig. 8.

We added these methods and description to the new Supplementary text: please see P. 40, L. 32–P. 41, L. 16 of new Supplementary Information.

Also, the aminoacylated Cys-tRNA is by convention written Cys-tRNA^{Cys}, which is different from the non-aminoacylated Cys-tRNA expressed simply as Cys-tRNA, for example. Thank you for calling our attention to the terminology.

Comment #8:

6. The 34S transulfidation experiment of sup fig 12 uis hard to follow and is not carefully explained

Response:

The method and results shown in Supplementary Fig. 11 are described in its legend, which may not be easily understood. We provide an extensive description of the protocol and an interpretation of results in the new text: P. 5, L. 14 – 20, P. 14, L. 10 – 16 and new Supplementary information text: P. 41, L. 24 – 29 and Supplementary Fig. 11 legend. In brief, we performed a stable isotope (^{34}S) tracer analysis of CysSSH formation from CysSH (sulfur transfer reaction), as catalyzed by EcCARS. To clarify the molecular mechanism of formation of CysSSH and CysSSSH from cysteine,

³⁴S-labeled L-cysteine (Cys-³⁴SH) was reacted with EcCARS, after which the mixtures were treated with HPE-IAM, followed by LC-ESI-MS/MS. We thus clarified that CARS catalyzes the reaction to produce CysS-(S)_n-H from cysteine (CysSH), when the sulfur is cleaved from the donor cysteine and transferred to the acceptor cysteine thiol to form CysS-(S)_n-H.

Comment #9:

4. The experiments in 4 h, e, j would indicate that CSE and CBS have an important contribution in persulfide formation? Which would contradict the overall message of the paper.

Response:

The same question was raised by Reviewer #3 (Comment #4), who actually asked us to perform a gene silencing study for CSE/CBS. The results clearly showed that CysSSH production that was apparently regulated by CSE/CBS depended solely on CARS2 expression in HEK293T cells (new Supplementary Fig. 18). In other words, CARS2 likely contributes to CysSSH (low-molecular-weight) production in the whole cell. We verified this idea *in vivo* by using *Cars2* KO mice that we just developed, as mentioned above: Please see new Fig. 6 and new Supplementary Fig. 20 and their legends. Please see P. 8, L. 5 – 13.

Also, Reviewer #1 noted an issue closely related to this comment but focusing more on the enzyme kinetics of CARS vs. CSE/CBS. Please see our response to Reviewer #1, comment #5, which again supports the predominance of CARS over CSE/CBS.

In addition to our alterations just mentioned and described in this response to the Reviewers, we made a few passages of modifications to improve the content of our work. These changes include the following:

1. We originally included 6 main figures but now have 10; we added 3 new figures, including, for example, data on *Cars2* KO mice, and we somewhat changed the order of their presentation.
2. We transferred certain main figures to the Supplementary Information (new Supplementary Figs. 7a, 24b, 25b), as well as deleted ARS and MeHg data (old Figs. 4h-j, Supplementary Figs. 14, 24, and their related descriptions).
3. We modified and improved the scientific and illustrative quality of the schematics (e.g., in new Figs. 3a, 4, 8, and 10), according to revisions that we made to respond to the Reviewers' comments.
4. The scientific validity and reproducibility of some original experiments were qualified rigorously by repeating these studies: please see new Figs. 2a,e and 4c,g; new Supplementary Fig. 12a; new Supplementary Table 3.
5. The experiments with CARS2 KO cells and their results and interpretation were explained and interpreted in the revised text much more carefully and in detail than in the old text (new Figs. 4 and 8; new Supplementary Fig. 15; P. 7, L. 13 – P. 8, L. 4).

We also made minor modifications in the main text to clarify our message. Please use the marked-up versions to check our revision.

We are most grateful for this careful review and thoughtful comments about our paper, which indeed helped us improve our work.

Reply to Reviewer #3

Reviewer #3 (Remarks to the Author):

General Comment:

Unlike reactive oxygen and nitrogen species the role of reactive sulfur species in cellular signaling, metabolism and redox homeostasis is underappreciated. The manuscript by Akaike et. al. entitled “Aminoacyl-tRNA synthetases govern protein polysulfidation and mitochondrial bioenergetics” shows that aminoacyl-tRNA synthetases (ARS) have persulfide synthase activity. The authors identified CARS2 (cysteinyl-tRNA synthetase) as a major contributor of protein polysulfidation in the cell. The modified amino acids are incorporated into de novo synthesized protein nascent chains. The authors aimed to uncouple the catalytic function in tRNA aminoacylation from the production of cysteine persulfidation. Since CARS2 is a mitochondrial enzyme the authors went on to describe several mitochondrial deficiencies caused by the lack of this activity. The concept of CARS being responsible for cys persulfide activity and sulfidation being introduced co-translationally to proteins is of high interest. However, some of the findings or their interpretations in the manuscript are hard to reconcile with the basic cell biology knowledge.

Response:

We are very grateful for this Reviewer’s keen insight into our present work on CARS and reactive persulfides.

To verify the idea that CARS2 truly functions as a major source of endogenous reactive persulfides, we sought to develop *Cars2* knockout (KO) mice and only recently succeeded in doing so in our laboratory during the revision of our original work. We developed two clones (lines) of *Cars2* KO mice, which allowed us to produce an *in vivo* model showing a very clear phenotype in terms of CARS2-dependent persulfide production in mice. We therefore decided to incorporate and highlight the *in vivo* results that we obtained with these *Cars2* KO mice (please see new Figs. 5-7 and new Supplementary Fig. 19-21 and their legends: P. 2, L. 8 – 11; P. 8, L. 14 – P. 9, L. 28; P. 16, L. 37 – P. 17, L. 20).

However, our original paper may have contained ambiguous and incomplete explanations, and we edited our paper to present a more consistent interpretation that is justified by our original data and new data that we added. We therefore believe that the present revision will address all concerns raised by this Reviewer.

Comment #1:

First, the mitochondrial protein synthesis system produces just a few proteins (13 in human), whereas the mitochondrial proteome is estimated to be more than 1000 proteins, but they are all synthesized in the cytosol and subsequently transported. CARS2 as a mitochondrial protein (the references are missing here) should only modify these 13 proteins synthesized in mitochondria and not the other mitochondrial proteins or cellular proteins. Thus, it is very hard to understand how a mitochondrial aminoacyl-tRNA synthetase can lead to persulfidation of the large part of the mitochondrial or cellular proteome and not only these 13 mitochondrial translation products if they are doing so co-translationally. What would be the scenario behind it? What is the location of CARS2? What is the cellular distribution of sulfidated proteins (at least mitochondria vs the rest of the cell)? How does the lack CARS1 and 2 affect cellular sulfidation in terms of localization?

Response:

In response to this Reviewer, we incorporated appropriate references in the revision: Refs. 28-30. We now also show experimentally and clearly via the Figure below

(provided here, only for this answer) that CARS2 is localized in the mitochondria and CARS1 is localized in the cytoplasm of HEK293T cells.

Figure. Intracellular localization of CARS1 and CARS2 in HEK293T cells. Immunofluorescence microscopy of HEK293T cells clearly shows the localization of CARS2 in mitochondria (middle panels). CARS2 co-localized with TOMM20 (translocase of outer mitochondrial membrane 20) (merge).

Indeed, CARS2 is responsible for translation of only the 13 mitochondrial proteins, not for translation and biosynthesis of all cell cytosolic proteins. Therefore, CARS2 is not a major contributor to cytoplasmic protein polysulfidation. However, we showed here that CARS2 contributes to almost 70% (or more) of total intracellular CysSSH formation, as evidenced by LC-MS/MS analysis with *CARS2* KO HEK293T cells (new Fig. 4) and *Cars2* KO mice (new Fig. 6 and Supplementary Fig. 20) as well. More important, formation of 20-30% of CysSSH in all cell proteins (polysulfidation) depended on CARS2 expression not only in the *in vitro* cell culture study but also in the *in vivo* experiment using *Cars2* KO mice, as evidenced by LC-MS/MS analysis with HPE-IAM (new Fig. 7). Because CysSSH can serve as a good substrate for CARS-catalyzed CysSSH-tRNA^{CysSSH} production (new Supplementary Fig. 8), CysSSH may be involved in nascent peptide biosynthesis catalyzed by CARS1 in the cells. As already discussed in the original text, protein Cys polysulfides are degraded by thioredoxin (Trx) and Trx reductase or chemically by particular electrophilic substances such as methylmercury (MeHg) that mediates the de-polysulfidation reaction. In addition, we reported that persulfide can be transferred and incorporated (from CysSSH) into Cys residues in proteins and cause polysulfidation via a post-translational modification (PTM) (Ref. 3), which indeed can occur with Drp1 associated with mitochondria in a manner depending on CARS2 expression (Figs. 7, 8d, e, and 10).

With regard to the localization of CARS2, we just obtained another important finding that clearly indicates that CysSSH is very effectively released from mitochondria into the extramitochondrial lumen, i.e., the cytoplasmic milieu, as shown in our revision (please see new Supplementary Fig. 22). This striking finding strongly supports our interpretation that CARS2 produces at least 70% or more of all intracellular CysSSH (free form), which is initially generated in mitochondria and is

released into the cytoplasmic lumen, and that CARS2 can partly sustain protein polysulfidation in both co-translational and PTM processes, as illustrated in new Figs. 7c, 8e, and 10a.

We added all related statements to the new next: P. 7, L.10 – 11; P. 9, L. 9 – 28

Comments #2:

Furthermore, the way to use various mutants defective in persulfidation vs aminoacylation that are introduced to CARS2-KO is a good direction based on their differential activities determined in vitro for a bacterial system and partially confirmed in the cellular system. However, a major question that remains to be determined experimentally is what are the effects on mitochondrial translation. The mutants specific to sulfidation activity should suppress the translation defect of KO to a wt level.

Response:

We should note again that *Cars2* KO mice have a clear phenotype showing remarkable suppression of production of CysSSH and other related sulfide derivatives (please see new Figs. 5-7 and new Supplementary Fig. 19-21 and their related information on P. 8, L. 14 – P. 9, L. 28), which provides strong *in vivo* evidence that CARS2 indeed functions as a major CysSSH-producing enzyme.

As this Reviewer will see in results in new Fig. 4, CARS2 can express translation-protein synthetic activities, regardless of persulfide-producing potential, as confirmed by our study with various CARS2 Lys mutants added back to *CARS2* KO HEK293T cells, and even *in vivo* in mice, i.e., *Cars2* heterozygote KO mice with intact translational activity but markedly reduced CysSSH production, as new Fig. 6 and new Supplementary Fig. 20 show (please also see - P. 8, L. 14 – P. 9, L. 28).

Such robust and convincing findings including *in vivo* data therefore indicate that CARS2 is truly functionally active as a major CPERS (regardless of its original CARS function), with this function being primarily responsible for endogenous persulfide (polysulfide) production in a manner independent of mitochondrial translational activity. We now discuss this issue in the new text: P. 2, L. 8 – 11; P. 8, L. 14 – P. 9, L. 28; P. 12, L. 5 – 25; P. 14, L. 33 – P. 15, L. 13.

Comments #3:

Along the lines, if CARS2 would be modifying nascent chains by sulfidation during their synthesis being a major contributor to the global cellular persulfidation, it is also critical to check the efficiency of cytosolic translation in CARS2 vs CARS1 deficient cells.

Without thorough investigation of the points above followed by careful interpretations the manuscript (at least its biological part) seem to be much too preliminary.

Response:

This comment is related to Comment #1 above, to which we already responded very carefully, with our response in fact being supported by our original and newly provided experimental data. We thus believe that our proposed idea is justified by our findings both biologically and scientifically.

[Other points]

Comments #4-1

1. To convincingly show that the cysteinyl-tRNA synthetase is a major contributor to protein persulfidation, a direct comparison of the amount of endogenous modified

proteins dependent on cystathionase and cystathionine β -synthase or the cysteinyl-tRNA synthetase would be helpful (even using down-regulation approaches).

Response:

The same question was raised by Reviewer #2 (Comment #9), and besides, according to your suggestion, we performed a gene silencing study for CSE/CBS. The results clearly showed that CysSSH production that was apparently regulated by CSE/CBS depended solely on CARS2 expression in HEK293T cells (new Supplementary Fig. 18). More specifically, knocking down CSE/CBS in HEK293T cells attenuated CysSSH production, which was observed only in wild-type cells and was not observed appreciably in *CARS2* KO cells. Such a decrease in CysSSH production evident in HEK293T cells correlated with a decrease in the intracellular concentration of cysteine, which indicated that CSE/CBS seem to be indirectly involved in CysSSH production, just by supplying Cys, a substrate for CARS2 to produce CysSSH. Please see P. 8, L. 5 – 13.

Our *in vivo* study with *Cars2* KO mice that we just developed, as mentioned above, supported this result. Our study also showed a lack of an apparent contribution of CSE/CBS and even 3-MST to CysSSH production in mice *in vivo*, at least under physiological conditions. Also, our investigations of the enzyme kinetics of CARS vs. CSE/CBS strongly suggest the predominance of CARS over CSE/CBS. Also, Reviewer #1 noted an issue closely related to this comment but focusing more on the enzyme kinetics of CARS vs. CSE/CBS. Please see our response to Reviewer #1, comment #5, which again supports the predominance of CARS over CSE/CBS. We added this conclusion, justified by our present experimental data, to the new text: P. 12, L. 5 – 25.

Comments #4-2:

2. Recent data of Coughlin et. al., JMG, 2015 already correlated mutations in CARS2 with deficiencies in respiratory chain complexes and associated mitochondrial disease in a patient study. Akaike and colleagues should acknowledge this study in their manuscript.

Response:

We noted this paper by Coughlin et al., which described a patient with combined heterozygous *CARS2* mutations, an in-frame glutamic acid deletion (Glu217del), and a substitution of proline to leucine (Pro251Leu). Because both mutations are located in highly conserved residues of the core catalytic domain, the mutations may cause conformational changes and instability of CARS2 protein. In fact, the CARS2 protein level and charged mt-Cys-tRNA^{Cys} amount were decreased in fibroblasts obtained from the patient with the *CARS2* mutation. Although these authors claimed in the paper that the clinical symptoms observed resulted from a loss of the canonical function of CARS2, the neurological disorders of patient may in fact be caused by impairment of the persulfide-producing activity of this mutant *CARS2*, if any, which actually overlapped with the observed dysfunction of aminoacylation of Cys-tRNA. We included comments about this finding in the new text: P. 12, L. 32 – P. 13, L. 5.

Comments #4-3:

3. Please, avoid bold statements such as the one in the abstract about evolution of the central dogma in molecular biology.

Response:

In response to this Reviewer's kind suggestion, we removed the statement "central dogma in molecular biology."

In addition to our alterations just mentioned and described in this response to the Reviewers, we made a few passages of modifications to improve the content of our work. These changes include the following:

In addition to our alterations just mentioned and described in this response to the Reviewers, we made a few passages of modifications to improve the content of our work. These changes include the following:

1. We originally included 6 main figures but now have 10; we added 3 new figures, including, for example, data on *Cars2* KO mice, and we somewhat changed the order of their presentation.
2. We transferred certain main figures to the Supplementary Information (new Supplementary Figs. 7a, 24b, 25b), as well as deleted ARS and MeHg data (old Figs. 4h-j, Supplementary Figs. 14, 24. and their related descriptions).
3. We modified and improved the scientific and illustrative quality of the schematics (e.g., in new Figs. 3a, 4, 8, and 10), according to revisions that we made to respond to the Reviewers' comments.
4. The scientific validity and reproducibility of some original experiments were qualified rigorously by repeating these studies: please see new Figs. 2a,e and 4c,g; new Supplementary Fig. 12a; new Supplementary Table 3.
5. The experiments with *CARS2* KO cells and their results and interpretation were explained and interpreted in the revised text much more carefully and in detail than in the old text (new Figs. 4 and 8; new Supplementary Fig. 15; P. 7, L. 13 – P. 8, L. 4).

We also made minor modifications in the main text to clarify our message. Please use the marked-up versions to check our revision.

Many thanks again for this careful review and insightful comments on our paper, which indeed helped us improve our work.

Reviewers' comments:

Reviewer #1 (Remarks to the Author):

This is a revised manuscript by Akaike et al demonstrating a novel role of CARS2 in the production in cysteine persulfates, and their potential role in eukaryotic biology.

The manuscript in its original submission represented a very comprehensive study, but the three reviewers nonetheless identified a number of concerning issues, ranging from the basic chemistry of Cys hydropersulfides, the specific mechanism of CARS Cys-SSH production, the relative contributions of CARS and CSE/CBE, and the extent of Cys hydropersulfide modification of the proteome.

The two noteworthy characteristics of this revision are (1) the great care and attention to detail that went into the responses to the critiques of all three original reviewers, and (2) the inclusion of new data based on the construction of CARS2 +/- heterozygous mice (two independent lines no less!). Taking (1) and (2) into account, the authors re-wrote large sections of the manuscript and eliminated some tangential findings from the initial submission that were not critical to the arguments concerning CARS2 function.

I believe that the authors have made an extremely compelling case for their surprising findings, and that no further modifications of the text or data are required or warranted.

Reviewer #2 (Remarks to the Author):

General comments

This is a revised paper. Authors have done a very good job at responding at most the enquiries and questions that were raised in the first evaluation. Nevertheless, a number of points have not been satisfactorily answered as described below. Despite this, although complex, this is an interesting study that deserves publication.

Comment #2:

Major point:

i) It is not clearly established that the reaction of polysulfides with halogens is an electrophilic attack, this reaction may proceed by oxidation or radical chemistry, ii) the mechanism of the reaction of iodoacetamide on thiocystine is unclear as the products reported in Ref. 26 are not those expected for an electrophilic attack. The expected products of an electrophilic attack are carboxymethylthiocysteine and formally a sulfenium cation (Cy-S⁺) as a leaving group which is

unstable and thus will react with a nucleophile, most probably water, leading to formation of sulfenic acid, but this compound is not detected or not reported in Ref. 26. Thereby, although carboxymethylthiocysteine was detected in this study, the presence of carboxymethylcysteine indicates that a Cys-thiolate (Cy-S⁻) anion is formed which is not consistent with the electrophilic mechanism. It is not unlikely that traces of iodide (I⁻) may catalyse the decomposition of thiocystine into cysteine and cysteine persulfide that in turn will amplify the decomposition of thiocystine and finally cysteine and cysteine persulfide are expected to rapidly react with iodoacetamide to yield carboxymethylcysteine and carboxymethylthiocysteine, respectively.

In conclusion, it is not clear how the reactions with the electrophiles described herein (Supp Fig. 1, 2 & 4 and afterwards), proceed and thus whether these reactions are relevant to detect protein bound hydropolysulfides. Although, Supp. Fig. 2 focus on hydropolysulfides which is more relevant than alkylpolysulfides for the analysis of polypersulfurated proteins *in vivo*, the identification of the products of the reaction is lacking and thus a clear conclusion on the polysulfidation of proteins cannot be drawn from these assays. To convince the reviewer and the reader, the authors must show ESI spectra of whole ADH5 and GAPDH proteins that should display several peaks corresponding to the persulfurated forms of the proteins.

Minor point

Providing some of the MS spectra in addition to the LC chromatograms (Supp Fig. 5, 6, 7 and 8) will be informative to support the interpretation of the authors.

Comment #3:

Major point

As the reaction of electrophiles on “internal” sulfur is not clearly established (see major point to response to comment2), it is critical here to validate the electrophilic degradation/PMSA assay developed by the authors. First by providing a compelling analysis of the product of the reaction and second by providing ESI spectra of the whole ADH5 and GAPDH recombinant proteins. A comparison of GAPDH and ADH5 treated with NEM as performed in sup Fig. 6 and 7 with HPE-IAM will further help strengthen these data. Also, heating a sample that contains alkylating agents promotes alkylation on non-cys residues, essentially histidine (as described in sup. Methods). This must be worked out by omitting the heating step for the samples presented in sup. Fig. 4.

Comment #4:

The authors have probably misinterpreted ref. 14. In this report (Ref. 14), polysulfidation is a consequence of the defect in Fe-S cluster biogenesis; in standard conditions, the cysteines coordinating the metal ion are not persulfidated, i.e. the Fe-S cluster is incorporated in proteins that are not persulfidated. This suggest that polysulfidation may occur only under stress

conditions with immature proteins, inactivated forms or misfolded proteins. It is moreover surprising that polysulfidation of ADH5 could preserve zinc binding. The corollary question is whether the persulfidated protein is still active?

Comment #6:

Major point

The reviewer thanks the authors for all these mechanistic details on PLP enzymes, but the question was whether or not the enzyme already contains PLP, which must be shown by providing a UV-visible spectrum of EcCARS as a peak is expected near 420 nm and quantification of PLP using specific absorption coefficient (ϵ is usually in the range of 10,000 M⁻¹.cm⁻¹ at 420 nm). The slight increase in Cys-SSH upon incubation with PLP must be correlated with increased PLP binding to EcCARS (to be assessed by UV-visible) (Fig. 2A). It is also surprising that human CARS2 must be reconstituted with PLP, as PLP is usually present in native proteins. PLP alone should be tested.

Comment #9:

Major point

The reviewer carefully examined this Fig. but could not find “the depended solely on CARS2” effect, since silencing of CBS or CSE and CARS2 KO decrease the global level of Cys-SSH to a similar extent. This assertion is maybe true for GSSH and HS- that are not or moderately affected by the silencing of CBS or CSE. The conclusion here must be clarified.

Reviewer #3 (Remarks to the Author):

The authors adequately addressed my criticism. This is a very important and interesting story.

Reply to the Reviewer #2's comments:

Reviewer #2 (Remarks to the Author):

General comments:

This is a revised paper. Authors have done a very good job at responding at most the enquiries and questions that were raised in the first evaluation. Nevertheless, a number of points have not been satisfactorily answered as described below. Despite this, although complex, this is an interesting study that deserves publication.

Response:

We are grateful for this reviewer's thoughtful and positive review of our revised manuscript. We responded to all five comments (#2, 3, 4, 6 & 9) from this reviewer, and performed very careful revision (please see the item-by-item statements below).

Comment #2:

Major point:

i) It is not clearly established that the reaction of polysulfides with halogens is an electrophilic attack, this reaction may proceed by oxidation or radical chemistry, ii) the mechanism of the reaction of iodoacetamide on thiocystine is unclear as the products reported in Ref. 26 are not those expected for an electrophilic attack. The expected products of an electrophilic attack are carboxymethylthiocystine and formally a sulfenium cation (Cy-S⁺) as a leaving group which is unstable and thus will react with a nucleophile, most probably water, leading to formation of sulfenic acid, but this compound is not detected or not reported in Ref. 26. Thereby, although carboxymethylthiocystine was detected in this study, the presence of carboxymethylcystine indicates that a Cys-thiolate (Cy-S⁻) anion is formed which is not consistent with the electrophilic mechanism. It is not unlikely that traces of iodide (I⁻) may catalyse the decomposition of thiocystine into cysteine and cysteine persulfide that in turn will amplify the decomposition of thiocystine and finally cysteine and cysteine persulfide are expected to rapidly react with iodoacetamide to yield carboxymethylcystine and carboxymethylthiocystine, respectively.

In conclusion, it is not clear how the reactions with the electrophiles described herein (Supp Fig. 1, 2 & 4 and afterwards), proceed and thus whether these reactions are relevant to detect protein bound hydropolysulfides. Although, Supp. Fig. 2 focus on hydropolysulfides which is more relevant than alkylpolysulfides for the analysis of polypersulfurated proteins in vivo, the identification of the products of the reaction is lacking and thus a clear conclusion on the polysulfidation of proteins cannot be drawn from these assays. To convince the reviewer and the reader, the authors must show ESI spectra of whole ADH5 and GAPDH proteins that should display several peaks corresponding to the persulfurated forms of the proteins.

Response:

This reviewer is slightly mistaken in interpreting of the results discussed in new reference 27 (old Ref. 26) [Abdolrasulnia and Wood, 1980]. These investigators were able to identify (via chromatography) carboxymethylthiocystine from the reaction of iodoacetate with Cys-SSS-Cys. They specifically indicate that they were unable to detect carboxymethylcystine. The paper states "Carboxymethylcystine was not detected as a product of the reaction because it has the same electrophoretic mobility as a carboxymethylthiocystine. It can be distinguished from the latter by a lack of a disulfide reaction with cyanide and nitroprusside reagent". This reviewer may have been confused by Eqn. 7 in the text which (for some reason) shows both carboxymethylthiocystine and carboxymethylcystine as products (even though carboxymethylthiocystine is the only fully characterized product discussed). Regardless, the expected product of a nucleophilic attack of a polysulfide on an established electrophile, iodoacetamide, is generated, carboxymethylthiocystine, again, consistent with the nucleophilic reactivity of a polysulfide.

The electrophilic reaction of halogens or iodoacetate with cystine trisulfide described in new Ref. 25-27 (old Ref. 24-26) were referenced here to simply provide evidence that the nucleophilic properties of polysulfides have been alluded to previously. To be sure, the mechanistic details of these reactions of polysulfides have not been firmly established

(as the reviewer accurately points out). Nevertheless, there appears to be ample evidence to support that polysulfur compounds (RSS_nR , $n > 2$) are nucleophilic (albeit not nearly as nucleophilic as terminal thiolate anions), for example, as verified experimentally herein and can be cleaved by electrophiles indeed (as evidenced by extensive decomposition by a strong electrophile NEM not by a weak electrophile like IAM; Supplementary Fig. 1). As further evidence for this process and to provide a little more mechanistic insight, we have added the reference to a review by Kice who discusses the cleavage of disulfides with electrophiles [Kice, JL (1968) Electrophilic and nucleophilic catalysis of the scission of the sulfur-sulfur bond, *Acc. Chem Res.*, 1, 58-64 (new Ref. 28)]. It is expected that the mechanism of cleavage of S-S bonds in polysulfides via reaction with electrophiles is exactly analogous to that proposed in this paper.

Also, as this reviewer mentioned, if an electron transfer reaction occurs, rather than a concerted electrophilic reaction, both pathways would be predicted to give the same product. A mechanistic rationale for this statement can be schematically depicted as follows:

Although the above scheme serves to explain the predicted (and observed) products of the electrophilic cleavage of S-S bonds, inclusion of this is, at this point, beyond the scope of this manuscript, which is already “dense” (as reviewers have pointed out). Future work from this lab will focus on the mechanisms of these reactions and these processes will be discussed in future publications. Regardless, it appears clear that electrophilic cleavage of S-S bonds is not only possible but expected in the experiments described in this manuscript. So, I am afraid that further argument may not produce any meaningful outcome, and needs massive chemical studies to be conducted carefully and rigorously in the future (not in our current paper), which is of course beyond the scope of this work.

Nevertheless, at the request of reviewer #2, we now show here in **Fig. 1** robust evidence revealing again that the cysteine hydropolysulfide formation is abundant and

prevalent in proteins generated co-translationally. Indeed, we thank this reviewer for this suggestion. We clearly identified high degrees of polysulfidation occurring at the ²⁷⁴Cys residue of the mature GAPDH protein expressed and synthesized in *E. coli*. As soon as the recombinant GAPDH was isolated from *E. coli*, without treatment of any alkylating agent or electrophile like IAM or HPE-IAM, followed by quick digestion with trypsin, which was promptly subjected to the LC-ESI-Q-TOF analysis, in a similar manner as shown for the PUNCH-PsP method (demonstrated in main Fig. 1c and Supplementary Fig. 10 in the earlier version of our manuscript). All native forms of CysSH, CysSSH, and CySSSH residues were efficiently recovered from the native whole GAPDH protein and the extension of polysulfidation reached more than 60% of the ²⁷⁴Cys residue of mature protein. Please note that even CysSH/CysSSH/CySSSH residues in the nascent peptides of GAPDH as detected by the PUNCH-PsP in the main Fig. 1c was consistently identified as their native SH forms without IAM labeling. All these rigorous LC-Q-TOF analyses unambiguously revealed that extensive and prevalent cysteine polysulfidation is introduced co-translationally and sustained in the mature protein physiologically present even in the post-translational processes of the cells.

Fig. 1. Direct identification by LC-Q-TOF of native forms of CysSH, CysSSH, and CySSSH residues present in mature GAPDH protein. The lower panel shows the same CysSH/CysSSH/CySSSH residues in the nascent peptides of GAPDH as detected by the PUNCH-PsP (main Fig. 1c in the earlier version of our manuscript).

In addition, this reviewer mentions that oxidation by electrophilic halogens (e.g.,

Br₂) can occur by oxidation (regarding a reference used to support electrophilic cleavage of S-S bonds). We absolutely agree. Mechanistically it likely occurs via electrophile-nucleophile chemistry as the initial step but regardless it is a formal oxidation no matter what the mechanism is. This is akin to the oxidation of thiols with acidic hydrogen peroxide or a peracid (this is an oxidation of sulfur that occurs via nucleophilic attack of the sulfur atom on an electrophilic oxygen atom of hydrogen peroxide or peracid to make the sulfenic acid). Indeed, the chemistry we propose for electrophilic S-S bond cleavage, is as well a formal oxidation, consistent with the chemical description of Br₂-mediated S-S bond cleavage.

This reviewer might be suspicious about artifactual polysulfidation that may be caused by sulfur oxidation that might be catalyzed by I⁻ released from IAM after reaction with sulfides, especially CysS⁻. In response to this, we have not considered this since we never observed any IAM-induced amplification reactions with low-molecular-weight polysulfides (e.g., cysteine and glutathione polysulfides) and even with whole native protein such as GAPDH as just described above. Granting that such artifacts induced by I⁻ potentially may occur, any extra sulfur species (extruded or existing endogenously) necessary for oxidative polysulfidation as substrates would have been present initially in each sample, for example, in GAPDH, ADH5, etc., tested in our present study. If this does occur, this is also consistent with the existence of polysulfides biosynthesized abundantly in various organisms.

In conclusion, the products we identified herein from the reaction with electrophiles are consistent with the idea we propose that protein-bound cysteine polysulfides are indeed present and prevalent. Nevertheless, we may have to further clarify the exact chemical mechanisms of the electrophilic polysulfide modification, which includes the fate of the extruded sulfur species after electrophilic reactions, the preferred site of electrophilic attacks, etc., in the next extended study. It is, however, hard to imagine that the observed products can result from anything else except polysulfide species, and thus it is quite logical to expect that these products must have come from polysulfides endogenously biosynthesized either directly or indirectly by CARs or CPERSs, and other enzymes like CSE etc.

The additional chemical insight into the electrophilic polysulfide modification is incorporated on P. 3, L. 37 – P. 4, L. 2; P. 14, L.10–14. Also, the direct evidence for the co-translational protein polysulfidation revealed by the LC-ESI-Q-TOF analysis is provided in the revised MS: P. 5, L. 15–22; Supplementary Fig. 10e, its legend, and its related information (P. 46).

Minor point

Providing some of the MS spectra in addition to the LC chromatograms (Supp Fig. 5, 6, 7 and 8) will be informative to support the interpretation of the authors.

Response:

Because the LC chromatograms (Supplementary Figs. 5, 6, and 8) are those of MS/MS analysis, no ESI spectra except the single *m/z* of MS/MS spectra is available. Meanwhile, some of the typical ESI spectra for Supplementary Fig. 7a, b, and 10d are now added (Supplementary information P. 10 and P. 14–15), according to this comment.

Comment #3:

Major point

As the reaction of electrophiles on “internal” sulfur is not clearly established (see major point to response to comment2), it is critical here to validate the electrophilic degradation/PMS assay developed by the authors. First by providing a compelling analysis of the product of the reaction and second by providing ESI spectra of the whole ADH5 and GAPDH recombinant proteins. A comparison of GAPDH and ADH5 treated with NEM as performed in sup Fig. 6 and 7 with HPE-IAM will further help strengthen these data. Also, heating a sample that contains alkylating agents promotes alkylation on

non-cys residues, essentially histidine (as described in sup. Methods). This must be worked out by omitting the heating step for the samples presented in sup. Fig. 4.

Response:

As demonstrated in **Fig. 1** above, the original forms of CysSSH and CysSSSH contained in GAPDH without IAM modification is clearly identified by this LC-ESI-Q-TOF analysis. We would like this reviewer to recognize that it is technically impossible to obtain truly accurate and reliable ESI-MS spectra of whole molecules of such high-molecular-weight proteins, e.g., GAPDH (36 kDa x 4, tetramer), ADH5 (40 kDa x 2, homodimer), etc., by using any types of mass spectrometers having precisely high mass resolution. In particular, we need to capitalize the high resolution ESI-MS, which can ensure high-performance, reproducible, and even quantitative MS analysis, to achieve any rigorous identification of polysulfidated cysteine moieties in the proteins. MADLI-TOF-MS is usually applicable for the protein MS analysis, but unfortunately, even the most current version of the instrument has no satisfactorily high performance required for measurements of protein-bound polysulfides.

For example, as shown in **Fig. 2** below (only for this review), an ESI spectrum of ETHE1, a relatively small 27 kDa protein, exhibits a single major peak accompanied by a few minor peaks, which appear to correspond to ETHE1 with different levels of polysulfidation with or without ferrous ion (ETHE1 is an iron-containing enzyme) based on $\Delta m/z$ from the estimated molecular weight of this recombinant ETHE1; albeit much precise and rigorous high-resolution mass identification is necessary to unequivocally reveal the presence of additional sulfur atoms in the protein.

Fig. 2. MALDI-TOF-MS analysis of recombinant human ETHE1 protein. Recombinant human ETHE1 highly purified by Ni-NTA agarose and gel filtration column (FPLC, Sephadex 200 Increase, GE Healthcare) was subjected to MALDI-TOF-MS analysis. Estimated molecular weight of ETHE1 was calculated with the Compute pI/Mw tool (http://www.expasy.ch/tools/pi_tool.html). The molecular mass of recombinant ETHE1 was determined using a DALTONICS Autoflex spectrometer (Bruker Japan) in the positive ion mode. Mass spectroscopic analysis was carried out in the linear mode using 3,5-dimethoxy-4-hydroxycinnamic acid as a matrix. In a typical run, protein solution was mixed with equal volume of a matrix solution (10 mg/ml) and air-dried on the sample plate for the MALDI-TOF-MS.

Nevertheless, we have succeeded to directly identify and quantify the CysSSH/SSSH

in GAPDH via our LC-ESI-Q-TOF by neatly utilizing the appropriate sizes of trypsin digests of the native GAPDH without electrophilic treatments (Fig. 1), as already discussed in the earlier response to comment #2.

Meanwhile, we showed here only a result of the Q-TOF analysis for GAPDH but not ADH5 or other proteins like ETHE1, because we wanted to extend the protein polysulfidation study to ADH5 and ETHE1, especially focusing on their functional relevance, which is already reported in part (new Ref. 15) and will become publishable soon in the future. For example, the polysulfidation of ETHE1 is clearly shown in new Ref. 15, and some new study with ETHE1 also unequivocally verified that a particular Cys residue of this protein is essential to maintain whole protein polysulfidation and thereby the catalytic function of this enzyme as a GSSH dioxygenase, as illustrated here in Fig. 3 and described in new Ref. 15. The same functional role in protein polysulfidation of a particular Cys residue is now identified with ADH5, and this most recent finding will be published elsewhere (just submitted to Sci Signal; please see a manuscript attached herewith).

Fig. 3. Protein polysulfidation of ETHE1 Cys-to-serine mutants. (a) Detection of protein polysulfidation in Cys-to-serine mutants. Protein polysulfidation of wild-type and Cys mutants of human ETHE1 was identified by using PMSA with ASBT and IAM. (b) Polysulfidation of ETHE1 proteins (wild type and C247S mutant) was identified via LC-ESI-MS/MS. Representative LC-ESI-MS/MS chromatograms of endogenous Cys and CysSSH residues of ETHE1, with isotope-labeled internal standards (left panel). The ratio CysSSH/Cys residue (% of wild type) is shown for each ETHE1 protein (right panel). Results are means \pm SD (n = 3). **P* < 0.01 (unpaired Student's *t*-test). Modified and updated from new Ref. 15.

Regarding the specificity of PMSA, especially for PEG-Mal labeling, we can show very clearly here in Fig. 3 (above) and Fig. 4 (below) the PMSA profiling eventually lacking the gel shift-up of various Cys-to-Ser mutants of ETHE1 and GAPDH. In brief,

the Cys mutations nullified the PEG-Mal labeling, resulting in loss of the gel shifts that are observed with the native WT proteins, which thus confirmed the specific labeling of PEG-Mal with Cys thiol and polysulfides, rather than other amino acid residues like His, Lys, and so on. We did not perform a control study as suggested by this reviewer, because the omission of the heat denaturing process impaired the quality of electrophoretic profile of protein bands on the gel. Nevertheless, the PMSA using Cys mutants should serve best as control proteins without any polysulfidation.

Fig. 4. PMSA profiles of GAPDH Cys-to-serine mutants. A scheme of this PMSA procedure (a), and the different PMSA profiles observed with various GAPDH Cys mutants. Please note that the degrees of gel shift-up are canceled as the number of Cys mutation is increased (as the Cys substitution is extended) from single mutation (C152S, C156S, C247S) to double (C152/156S, C156/247S) and triple (C152/156/247S) mutations.

Moreover, the treatment of ADH5 with NEM indeed completely abrogated the HPE-IAM labeling of CysSH and CysSSH/SSSH as evidenced by LC-ESI-MS/MS analysis shown in Fig. 5 below. This data thus indirectly support the electrophilic decomposition of protein-bound cysteine polysulfides induced by a strong electrophile NEM. Please see P. 4, L 25 – 29 in the revised main text and Supplementary Fig. 6b and its legend.

CysS-(S)_n-H in ADH5

Fig. 5. Effects of NEM treatment of the amounts of CysS-(S)_n-H detected in ADH5. The quantitative measurement of CysS-(S)_n-H formed in ADH5 was performed in a manner same as in the main Fig. 1a, except that 60 mM NEM was added to the pronase digest of ADH5 during the HPE-IAM labeling of CysS-(S)_n-H.

Comment #4:

The authors have probably misinterpreted ref. 14. In this report (Ref. 14), polysulfidation is a consequence of the defect in Fe-S cluster biogenesis; in standard conditions, the cysteines coordinating the metal ion are not persulfidated, i.e. the Fe-S cluster is incorporated in proteins that are not persulfidated. This suggests that polysulfidation may occur only under stress conditions with immature proteins, inactivated forms or misfolded proteins. It is moreover surprising that polysulfidation of ADH5 could preserve zinc binding. The corollary question is whether the persulfidated protein is still active?

Response:

Because of the indirect evidence we found recently that cysteine persulfide can contribute to the Fe-S clusters as a ligand for ferrous ion (new Ref. 14), we just speculated that the same mechanism might be involved in the polysulfidation (trisulfide bridge formation) reported in old Ref. 14. As this reviewer pointed out, however, at least the authors of this paper seem not to suppose such a polysulfidic iron complex potentially formed in the protein. We simply deleted this reference from the revised text, accordingly. The polysulfidation of particular Cys ligands for zinc ion in ADH5 and their functional consequences are discussed extensively in our separate study, which is just submitted to another journal as mentioned above (see an attached manuscript file).

Comment #6:

Major point

The reviewer thanks the authors for all these mechanistic details on PLP enzymes, but the question was whether or not the enzyme already contains PLP, which must be shown by providing a UV-visible spectrum of EcCARS as a peak is expected near 420 nm and quantification of PLP using specific absorption coefficient (ϵ is usually in the range of 10,000 M⁻¹.cm⁻¹ at 420 nm). The slight increase in Cys-SSH upon incubation with PLP must be correlated with increased PLP binding to EcCARS (to be assessed by UV-visible) (Fig. 2A). It is also surprising that human CARS2 must be reconstituted with PLP, as PLP is usually present in native proteins. PLP alone should be tested.

Response:

Yes, we certainly understand that we can assess the amount of PLP at 420 nm by UV-visible spectroscopy. This spectroscopic measurement is not precisely quantitative, however. We therefore decided to employ more quantitative and even sensitive approach by LC-ESI-MS/MS using 2,4-dinitrophenylhydrazine (DNPH) for labeling PLP, according to the method reported previously (PLoS One 6, e27643, 2011; Supplementary Ref. 17) with slight modifications. The DNPH-labeling LC-MS/MS analysis therefore indicated that the amounts of PLP bound to WT EcCARS and four different Lys mutants correlated well with their CPERS (persulfide producing) activities, as shown below in **Fig. 6a**. Very nice correlation was also found between the CPERS activity and PLP content of CARS2 containing varied amounts of PLP incorporated after treatment with different concentrations of PLP (see **Fig. 6b** below). No appreciable cysteine persulfide production was detected in the reaction mixture of cysteine and PLP alone as long as no more than 100 μM PLP was used. Please see the new Fig 3e, and new Supplementary Fig. 13b, their legends, and related statements (P. 6, L. 33–37; P. 7, L. 33–36; P. 24, L. 13–18; Supplementary Information P. 47, 2nd paragraph).

(New Supplementary Fig. 13b)

Main Fig. 2c

(New Main Fig. 3e)

Main Fig. 3d

Fig. 6. Precisely quantitative analysis for PLP bound to EcCARS and human CRAS2 and their correlation with CPERS activities. The mounts of PLP was quantified by LC-ESI-MS/MS analysis after PLP was extracted from each protein tested via the trapping reaction with DNPH. Either EcCARS or human CARS2 (15 μM each) was treated or untreated with various concentrations of PLP in 30 mM HEPES buffer (pH 7.5) at 37 $^{\circ}\text{C}$ for 1 h, after which free forms of PLP (protein unbound) were completely eliminated with the PD SpinTrap G-25 column, and the protein fraction recovered was further reacted with 2 mM DNPH to form PLP-DNPH adduct in 30 mM HEPES buffer (pH 7.5) at 37 $^{\circ}\text{C}$ for 1 h, followed by quantification by LC-ESI-MS/MS analysis. The CPERS activity of each protein was very well correlated with the amounts of PLP bound to the respective EcCARS and CARS2 proteins, including various EcCARS mutants, as shown in the right panels (main Fig. 2c and 3d in the earlier version of our manuscript).

Comment #9:

Major point

The reviewer carefully examined this Fig. but could not find “the depended solely on CARS2” effect, since silencing of CBS or CSE and CARS2 KO decrease the global level of Cys-SSH to a similar extent. This assertion is maybe true for GSSH and HS- that are not or moderately affected by the silencing of CBS or CSE. The conclusion here must be clarified.

Response:

We did not mention this phrase in the main text, but just described in the earlier version of Response to this reviewer #2’s Comment #9. Besides, our interpretation had been provided in the previous revision. Specifically, on P. 9, “These results suggest that CSE and CBS do not contribute directly to persulfide production but rather may promote the biosynthesis of cysteine and its supply to CARS, at least in this cultured cell model under physiological conditions”; and on P. 13, “Under physiological conditions, CSE and CBS seem to indirectly contribute to CysSSH production mainly by promoting the formation and supply of cysteine for CARS; this does not exclude the existence of other routes of polysulfide formation involving nitrosative and oxidative H₂S metabolism and intermediate hydropolysulfide generation. Also, CSE and CBS may still play a major role in the CysSSH production via the direct catalytic reaction using cystine as the substrate especially under pathophysiological conditions associated with oxidative and electrophilic stress, where intracellular cystine concentrations are considerably approaching the high K_m value of CSE”. Nevertheless, we explained again the results shown in the Supplementary Fig. 18, as follows.

As the Supplementary Fig. 18a clearly shows, the inhibitory effect of CSE and CBS by their gene silencing was most prevalently observed for CysSH production in both WT and CARS2 KO cells. On the contrary, while the level of CysSSH was suppressed in the WT cells after knockdown of CSE or CBS, the CSE/CBS knockdown did not affect the CysSSH production in CARS2 KO cells. These different profiles of CysSH and CysSSH productions affected by the CSE/CBS knockdown in the presence or absence of CARS2 expression clarified the following points (please see also **Fig. 7** below).

- 1) CysSH production is dependent on both CSE and CBS, and thus CysSH is provided via the metabolic pathways mediated by CSE/CBS in both cell lines irrespective of CARS2 expression.
- 2) Almost two thirds of CysSSH is supplied by CARS2 in HEK293T cells based on the decrease by almost two thirds in the CysSSH levels. The rest of CysSSH in the CARS2 KO cells were not derived from CSE/CBS expressed in HEK293T cells, since no further reduction of CysSSH was obtained even by CSE/CBS knockdown in CARS2 KO cells.
- 3) Therefore, it is logical to interpret that CysSSH is produced mainly by CARS2, but not by CSE/CBS in HEK293T cells. Nevertheless, it is highly plausible that CSE and CBS may be contributing to the CysSSH production by effective and sustaining supply of CysSH, a major substrate for the CysSSH biosynthesis catalyzed by CARS2 in HEK293T cells. This notion may be supported by the inversed increase in homocysteine content by silencing CSE/CBS. A scheme in **Fig. 7** below shows the major pathway for endogenous biosynthesis of cysteine as catalyzed by CSE and CBS from homocysteine and serine, and thereby supply the excellent substrate for CARS forming CysSSH, i.e., CPERS located in the most downstream of CSE/CBS.
- 4) Although the baseline polysulfide formation in HEK293T cells does not seem to be directly mediated by CSE or CBS under the currently applied physiological culture conditions, the posttranslational polysulfidation may be mediated by CSE or CBS, for example, under the oxidative stress, where the substrate cystine supply is elevated for CSE/CBS to produce substantial amounts of cysteine persulfide in a CARS-independent manner, via a collateral pathway illustrated with dash lines in the scheme below (**Fig. 7**).

The above explanation had been included as just mentioned (P. 9, L. 2–4; P. 13, L. 8 – 16) and somehow strengthened here in the main text of revised paper (P. 8, L. 32 – P. 9, L. 2).

Fig. 7. The metabolic pathway producing CysSH and CySSSH as catalyzed by CSE/CBS and by CARS/CPERS located in the most downstream.

In addition to the alterations mentioned above, we made a few passages of modifications, which include incorporation of our two new papers cited as new Ref. 5 & 6, which are related to the prevalent endogenous polysulfide formation we verified recently in humans, and the Ref. 6 pdf document is uploaded herewith. Also, three references (old Ref. 14, 37 & 39) have been removed, because old Ref. 37 & 39 are not directly related to the findings reported herein, and old Ref. 14 was deleted in response to the Comment #4. Please use the marked-up versions to check our revision.

Again, we would like to thank this reviewer for extremely careful review and comments.

Reviewer #2 (Remarks to the Author):

Authors have satisfactorily answered all the comments raised in the first and second revision, but one crucial point: The provided MALDI-TOF analysis of polysulfided proteins is very poor. We advised using ESI-MS, which was not performed: authors argue that this technique “cannot apply to such high molecular weight proteins, e.g., GAPDH (36 kDa x 4, tetramer), ADH5 (40 kDa x2, homodimer), etc., by using any types of mass spectrometers having precisely high mass resolution “.

We strongly disagree with authors: ESI-MS in the presence of acetonitrile should dissociate all non-covalent quaternary interactions, only leaving the covalent ones, as done in other studies. This is a central point of the all study and should be performed.

Reply to the Reviewer #2's comments:

Reviewer #2 (Remarks to the Author):

Authors have satisfactorily answered all the comments raised in the first and second revision, but one crucial point: The provided MALDI-TOF analysis of polysulfided proteins is very poor. We advised using ESI-MS, which was not performed: authors argue that this technique "cannot apply to such high molecular weight proteins, e.g., GAPDH (36 kDa x 4, tetramer), ADH5 (40 kDa x 2, homodimer), etc., by using any types of mass spectrometers having precisely high mass resolution ". We strongly disagree with authors: ESI-MS in the presence of acetonitrile should dissociate all non-covalent quaternary interactions, only leaving the covalent ones, as done in other studies. This is a central point of the all study and should be performed.

Response:

We are happy to learn that the reviewer is now satisfied with our analyses and feels that his/her concerns were generally adequately addressed.

Regarding the methodological comment, we agree that ESI-MS is also a capable technique to study the per/polysulfide formation on intact protein samples, for example, by capitalizing on the utility of multivalent ions for the ESI-MS analysis.

However, we feel that this issue is unrelated to the original point raised by the reviewer regarding an important aspect of this study, i.e. to provide convincing evidence for the proposed prominent presence of per/polysulfidation on protein Cys residues. We strongly believe that this concern is satisfactorily addressed by using 3 conceptually different techniques that are based on technically different mass spectrometry methods, as follows.

- 1) We developed a quantitative method to measure Cys polysulfide species in proteins after a pronase digestion step utilizing isotopically labeled internal standards. With the aid of this method we have shown excessive amounts of polysulfidation on GAPDH, ADH5 and ETHE1 Cys residues, and even on endogenous proteins in human cultured cells and mouse tissues *in vivo*.
- 2) After tryptic digestion, using ESI-MS detection we quantitatively demonstrated the polysulfidation of specific Cys residues in GAPDH and ADH5 proteins.
- 3) We used MALDI-TOF-based mass spectrometry to measure protein Cys polysulfidation of intact proteins (shown for ETHE1).

Therefore, although ESI-MS-based intact protein measurements (e.g., multivalent ion MS) could also potentially detect protein polysulfidation, we feel that we already have substantial and rigorous data to adequately support our conclusion that polysulfidation is a prominent modification on protein Cys residues.